# Pig genome functional annotation enhances the biological interpretation of complex traits and human disease

Zhangyuan Pan[1,9], Yuelin Yao[2,9], Hongwei Yin [3], Zexi Cai [4], Ying Wang[1], Lijing Bai[3], Colin Kern [1], Michelle Halstead [1], Ganrea Chanthavixay[1], Nares Trakooljul [5], Klaus Wimmers [5], Goutam Sahana[4], Guosheng Su[4], Mogens Sandø Lund[4], Merete Fredholm[6], Peter Karlskov-Mortensen [6], Catherine W. Ernst [7], Pablo Ross [1], Christopher K. Tuggle [8], Lingzhao Fang [2✉] & Huaijun Zhou [1✉]

The functional annotation of livestock genomes is crucial for understanding the molecular mechanisms that underpin complex traits of economic importance, adaptive evolution and comparative genomics. Here, we provide the most comprehensive catalogue to date of regulatory elements in the pig (*Sus scrofa*) by integrating 223 epigenomic and transcriptomic data sets, representing 14 biologically important tissues. We systematically describe the dynamic epigenetic landscape across tissues by functionally annotating 15 different chromatin states and defining their tissue-specific regulatory activities. We demonstrate that genomic variants associated with complex traits and adaptive evolution in pig are significantly enriched in active promoters and enhancers. Furthermore, we reveal distinct tissue-specific regulatory selection between Asian and European pig domestication processes. Compared with human and mouse epigenomes, we show that porcine regulatory elements are more conserved in DNA sequence, under both rapid and slow evolution, than those under neutral evolution across pig, mouse, and human. Finally, we provide biological insights on tissue-specific regulatory conservation, and by integrating 47 human genome-wide association studies, we demonstrate that, depending on the traits, mouse or pig might be more appropriate biomedical models for different complex traits and diseases.

[1] Department of Animal Science, University of California, Davis, Davis, CA, USA. [2] MRC Human Genetics Unit at the Institute of Genetics and Molecular Medicine, The University of Edinburgh, Edinburgh EH4 2XU, UK. [3] Agricultural Genome Institute at Shenzhen, Chinese Academy of Agricultural Sciences, 518120 Shenzhen, China. [4] Center for Quantitative Genetics and Genomics, Faculty of Technical Sciences, Aarhus University, Tjele 8300, Denmark. [5] Leibniz-Institute for Farm Animal Biology, Dummerstorf, Germany. [6] Animal Genetics, Bioinformatics and Breeding, Department of Veterinary and Animal Sciences, University of Copenhagen, Frederikgsberg C 1870, Denmark. [7] Department of Animal Science, Michigan State University, East Lansing, MI, USA. [8] Department of Animal Science, Iowa State University, Ames, IA, USA. [9] These authors contributed equally: Zhangyuan Pan, Yuelin Yao. ✉email: Lingzhao.fang@igmm.ed.ac.uk; hzhou@ucdavis.edu

Functional elements play essential roles in regulating gene expression in living cells and tissues[1]. There have been great efforts to identify and annotate functional elements in human and mouse genomes[1–11] as well as other model organisms, including *Drosophila*[12] and *C. elegans*[13]. Significant enrichment of variants associated with human complex traits within regulatory elements has demonstrated the importance of the resulting Encyclopedia of DNA Elements (ENCODE) data[14]. Comparative analysis of epigenomes and transcriptomes across species could provide novel insights into the molecular mechanisms underlying human disease[8,15]. Genetic variants associated with common illnesses are enriched in human orthologues of mouse regulatory elements identified by ENCODE[9], which suggests that the mouse could serve as a biomedical model for understanding some human diseases. However, compared with the mouse, pig (*Sus scrofa*) has more anatomical and physiological similarities to humans[16–18], and has been widely used as a human medical model[16,17,19–21]. The pig is also one of the most important farm animal species for meat production worldwide[22]. The genetic improvement of economically important complex traits such as growth, feed efficiency, and health could contribute to efficient and sustainable production of animal protein, contributing to a secure food supply for a growing world population. Causative variants associated with complex traits often have a small genetic effect on phenotypic variation, making them difficult to discover[23]. Functional annotation of regulatory elements in pig will lay a solid foundation for the identification of these causative variants, due to their enrichment in regulatory regions.

Following ENCODE and Roadmap Epigenomics projects[8], the Functional Annotation of Animal Genomes (FAANG) initiative[24], although still in its infancy, has made great progress towards annotating functional elements in many tissues across multiple domestic species, including pigs[25–30]. Here, we present 95 new genome-wide sequencing datasets from six gut-associated porcine tissues and integrate them with 128 previously published FAANG datasets from eight biologically distinct tissues. The collective interpretation of these datasets yields the most comprehensive annotation of functional elements to date in any domesticated animal species. In addition, we find that tissue-specific regulatory elements were enriched for the potential causative variants of complex phenotypes by integrating a variety of large-scale genome-wide association studies (GWAS) and expression quantitative trait loci (eQTL) datasets. Furthermore, by integrating signatures of selection in the pig genome, we show that tissue-specific regulatory elements likely played important roles in pig domestication. Finally, we compared porcine functional annotations with complementary datasets from the human and mouse, and integrated GWAS datasets concerning 47 human complex traits. These comparisons demonstrate conservation of tissue-specific epigenetic signatures, suggesting that, depending on the specific human diseases under investigation, either the pig or the mouse may be a more suitable animal model. Here, we show our systematic functional annotation of the pig genome significantly enhances our understanding of genetic control of complex traits in pigs and disease in humans.

## Results

### Data summary

We integrated 223 genome-wide sequencing datasets from 14 major tissues in pig (Fig. 1a), representing four histone modifications (H3K4me3, H3K4me1, H3K27ac, and H3K27me3) measured by Chromatin Immunoprecipitation sequencing (ChIP-seq), chromatin accessibility by the Assay for Transposase-Accessible Chromatin (ATAC-seq), DNA methylation by Reduced Representation Bisulfite sequencing (RRBS), and

gene expression by RNA-seq (Supplementary Fig. 1). We produced nearly 9 billion mapped reads with an average rate of 68.81% remaining after alignment and filtering across samples (Supplementary Data 1). Among 14 tissues, we obtained an average of 32,387, 106,849, 72,252, 98,721, and 122,585 peaks for H3K4me3, H3K4me1, H3K27ac, H3K27me3, and ATAC, with average size of 794, 1894, 618, 1190, and 653 bp, and covering 1.56, 2.78, 2.37, 7.74, and 3.31% of the entire genome, respectively (Fig. 1b, c and Supplementary Fig. 2). Additionally, we utilized 16 CTCF ChIP-seq datasets from eight tissues[29] and four Hi-C datasets from liver[30] to identify CTCF and Hi-C loops for associating regulatory elements (enhancers) with potential target genes.

The hierarchical clustering of samples based on the signal intensity of epigenetic marks and gene expression profiles clearly recapitulated sequencing assays, followed by tissue types and biological replicates (Fig. 1d), which was consistent with results of principal component analysis (PCA) (Supplementary Fig. 3). The six assays formed three major clusters: (1) active regulatory regions (H3K4me3, H3K27ac, H3K4me1, and ATAC), (2) Polycomb repression (H3K27me3), and (3) gene expression (RNA-seq). The four active regulatory marks were positively correlated with each other, but were negatively correlated with H3K27me3 – especially H3K27ac. The signal intensity of RNA-seq (within gene bodies) showed a weakly positive correlation with active regulatory marks, and a negative correlation with H3K27me3. Overall, three active regulatory marks (ATAC, H3K4me3, H3K27ac) showed a peak at the upstream of transcription start sites (TSS) of genes across tissues (Fig. 1e), whereas H3K4me1 showed a peak at 1 kb distance upstream of TSS (Fig. 1e).

To illustrate the complex interplays of regulatory elements and gene expression with respect to *Escherichia coli* infection and microvillar membrane morphology in intestinal tissues[31,32], we present an analysis of Myosin 1A (*MYO1A*). *MYO1A* is specifically and highly expressed in intestinal tissues and showed specific enrichment for H3K27ac signal around its TSS in intestinal tissues but not in other tissues (Fig. 1f). In addition, the TSS of *MYO1A* was accessible and enriched for other active regulatory marks (i.e., H3K27ac, H3K4me3, and H3K4me1) but not for Polycomb repression (H3K27me3) (Fig. 1f).

**Prediction and characterization of chromatin states across 14 tissues**. We defined 15 distinct chromatin states by combining all five epigenetic marks across 14 tissues. These states mainly represented promoters (TssA, TssAHet, and TssBiv, covering 1.16% of the entire genome), TSS-proximal transcribed regions (TxFlnk, TxFlnkWk, and TxFlnkHet, covering 0.92% of the genome), enhancers (EnhA, EnhAMe, EnhAWk, EnhAHet, and EnhPois, covering 6.5% of the genome), repressed regions (Repr and ReprWk, covering 13.25% of the genome), and quiescent regions (Qui, 73.39%) (Fig. 2a–e and Supplementary Data 2). Collectively, we identified 2,097,958 regulatory elements (excluding Qui) spanning 14 tissues, including 39,351 active promoters (TssA), 188,827 active strong enhancers (EnhA), and 142,821 repressors (Repr) (Supplementary Fig. 4a–c). On average, 4.79% of the genome was accessible but did not coincide with any other measured epigenetic marks (ATAC islands), indicating that additional epigenetic marks are required to further explore the biological function of such regions. TssA and TssBiv showed the highest enrichment of conserved DNA sequence elements, followed by TSS-proximal transcribed regions and accessible enhancers (EnhA and EnhAMe) (Fig. 2f). In general, TssA and TssBiv showed the highest enrichment at TSS, while other chromatin states showed enrichment both up- and down-stream

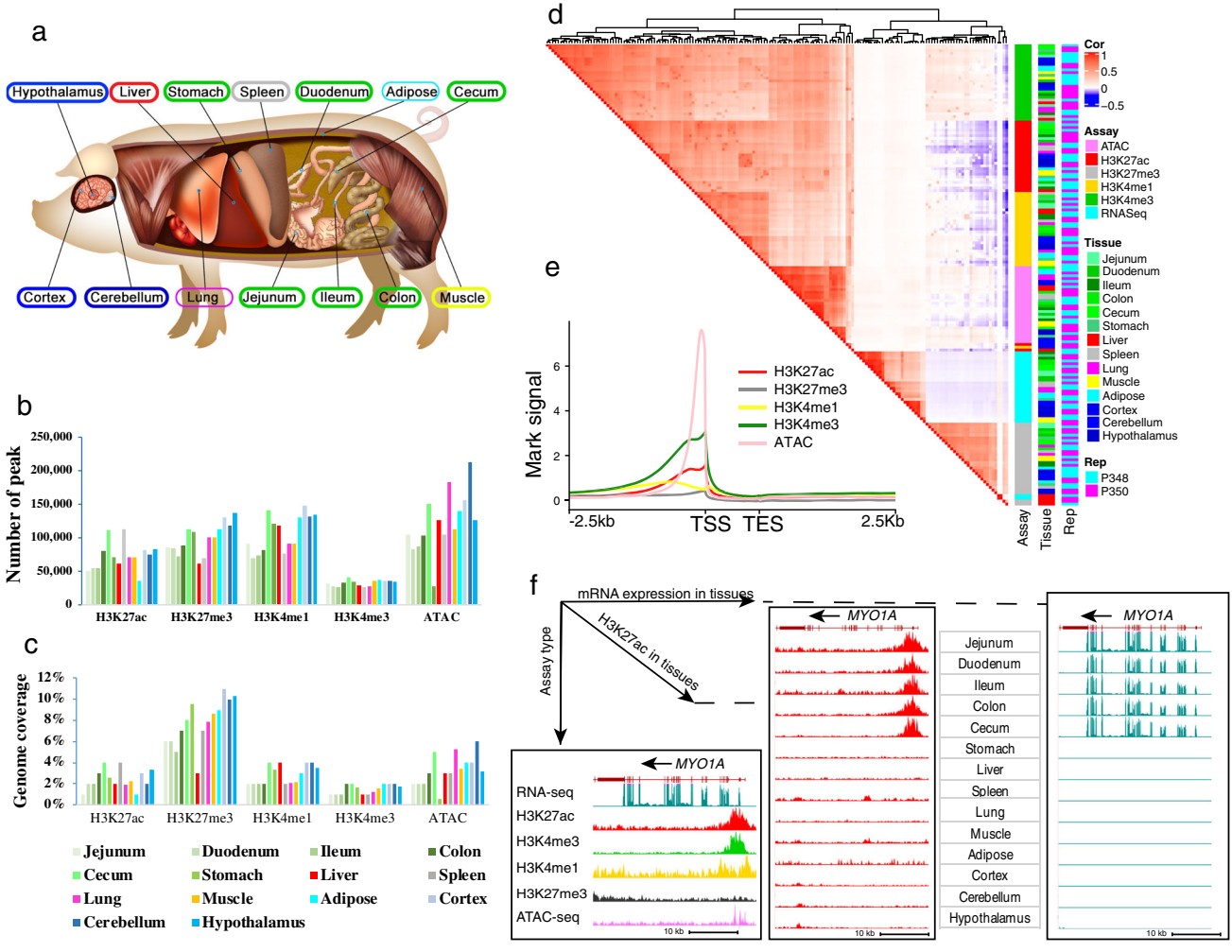

**Fig. 1 Data summary of epigenomic information across tissues and marks. a** Tissues assayed by this study. **b**, **c** Average peak number and genome coverage for each epigenetic mark in each tissue. **d** The Pearson correlations among assays, tissues, and biological replicates (P348 and P350) based on the normalized signal in 1 kb windows stepped across the whole genome. **e** Average epigenetic mark signal proximal to protein-coding genes. TSS transcription start site, TES transcription end site. **f** Epigenetic signal at the *MYO1A* locus according to different assays and in different tissues. Vertical scale of UCSC tracks shows normalized signal from 0 to 200 for RNA-seq, 0 to 100 for H3K27ac and H3K4me3, and 0 to 50 for other marks and ATAC-seq.

of TSS (Fig. 2g). For instance, TssAHet and TSS-proximal transcribed states had the highest enrichment ~2 kb upstream of TSS, whereas enhancer states showed the highest enrichment ~20 kb upstream of TSS. Repressed states were enriched ~20 kb up- and down-stream of TSS (Fig. 2g).

In general, different chromatin states showed distinct DNA methylation levels (Fig. 2h). Promoter and TSS-proximal transcribed states were hypomethylated compared to nearby sequences (10 kb up- and down-stream of TSS). Among promoter states, TssA had the lowest methylation level, confirming the well-known negative correlation between promoter methylation and gene expression[33]. The enhancer states showed intermediate methylation levels, among which EnhA and EnhAMe had lower methylation levels compared to other enhancers (Fig. 2g). In addition, we also observed that EnhA and EnhAMe had more conserved sequence than other enhancers (Fig. 2f). This result suggests accessible enhancers may have more conserved sequences than non-accessible enhancers.

To explore and illustrate the relationships among chromatin states, individual epigenetic marks, gene density, gene expression, DNA methylation, and chromatin conformation we focused chromosome 7 (Chr7) (Fig. 2i). We observed that regions with

higher density of genes were characterized by active chromatin states, higher gene expression, increased chromatin accessibility, and lower methylation levels. More chromatin interactions (measured by topologically associating domains (TADs) from Hi-C data) were detected within both gene deserts and gene rich regions than in the remaining Chr7 genomic regions. To examine the associations of chromatin states with gene expression across tissues, we investigated the *VIL1* locus (Villin-1), which participates in response to intestinal inflammation[34] (Fig. 2j). *VIL1* exhibited tissue-specific active promoters and enhancers, as well as high expression in intestinal tissues compared to other tissues. Of particular note, despite the presence of TssA, *VIL1* was not expressed in stomach, possibly due to the lack of enhancer activity upstream of its TSS, suggesting that enhancers, together with promoters, collectively regulate *VIL1* expression. These patterns were also observed for *MYO1A* and Hepatocyte Nuclear Factor 4 Gamma (*HNF4G*), a gene that plays key roles in enterocyte differentiation[35] and renewal of intestinal stem cells[36] (Supplementary Fig. 4).

**Dynamics of chromatin states across genome and tissues**. We clustered the entire genome into 12 modules based on their

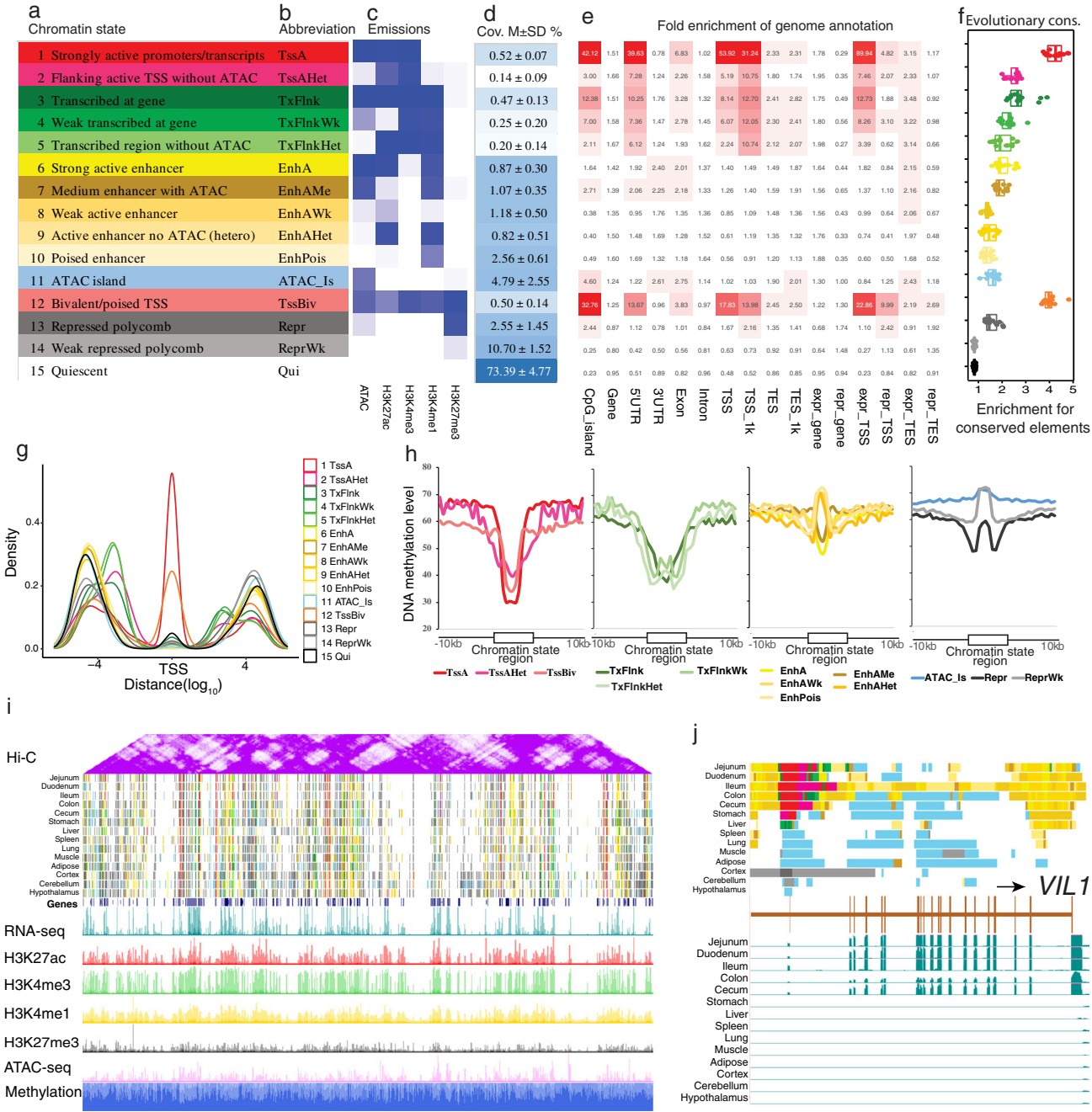

**Fig. 2 Chromatin landscape across 14 tissues. a**, **b** Definitions and abbreviations of 15 chromatin states. **c** Emission probabilities of individual epigenetic marks for each chromatin state. The color from white to deep blue indicates emission probability (0–1). **d** Genomic coverage of each chromatin state. M ± SD mean ± standard deviation. **e** Average enrichment of chromatin states for genomic annotations, including CpG islands, genes, TSS/TES_1K (±1 kb around TSS and TES), expressed genes (TPM ≥ 0.1), and repressed genes (TPM < 0.1) in each tissue. **f** Fold enrichments of chromatin states for non-coding mammalian conserved elements from Genomic Evolutionary Rate Profiling (GERP). Whiskers show 1.5× interquartile range. Each data point represents one of 14 different tissues. **g** Density of each chromatin state in positions relative to gene TSS. **h** Average methylation level of chromatin states in jejunum. **i** Hi-C (250 kb resolution), predicted chromatin states, epigenetic signal, and normalized methylation level in jejunum across chromosome 7. **j** Chromatin state landscape and mRNA expression at *VIL1* locus (chr15:120,459,825-120,493,312, susScr11) across 14 tissues. Vertical scale of UCSC tracks shows normalized signal from 0 to 200 for RNA-seq.

relative frequency of chromatin states and observed that these modules exhibited distinct enrichments for protein-coding genes, non-coding genes, and CpG islands (Fig. 3a). For instance, module 2 (M2) was characterized by active promoters and accessible enhancers, had the highest enrichment for genes and CpG islands, the lowest levels of DNA methylation, and the highest gene expression levels (Fig. 3b). Compared to M11 and

M12, M10 showed similar enrichment for Polycomb repression but higher enrichment for TssBiv, in which genes exhibited significantly lower expression levels, thereby suggesting the crucial role of TssBiv for regulating gene repression (Fig. 3b). In addition, we noticed that module 1 had high enrichment for TssAHet, high levels of DNA methylation, and high representation of genes located on the X chromosome, which were functionally enriched

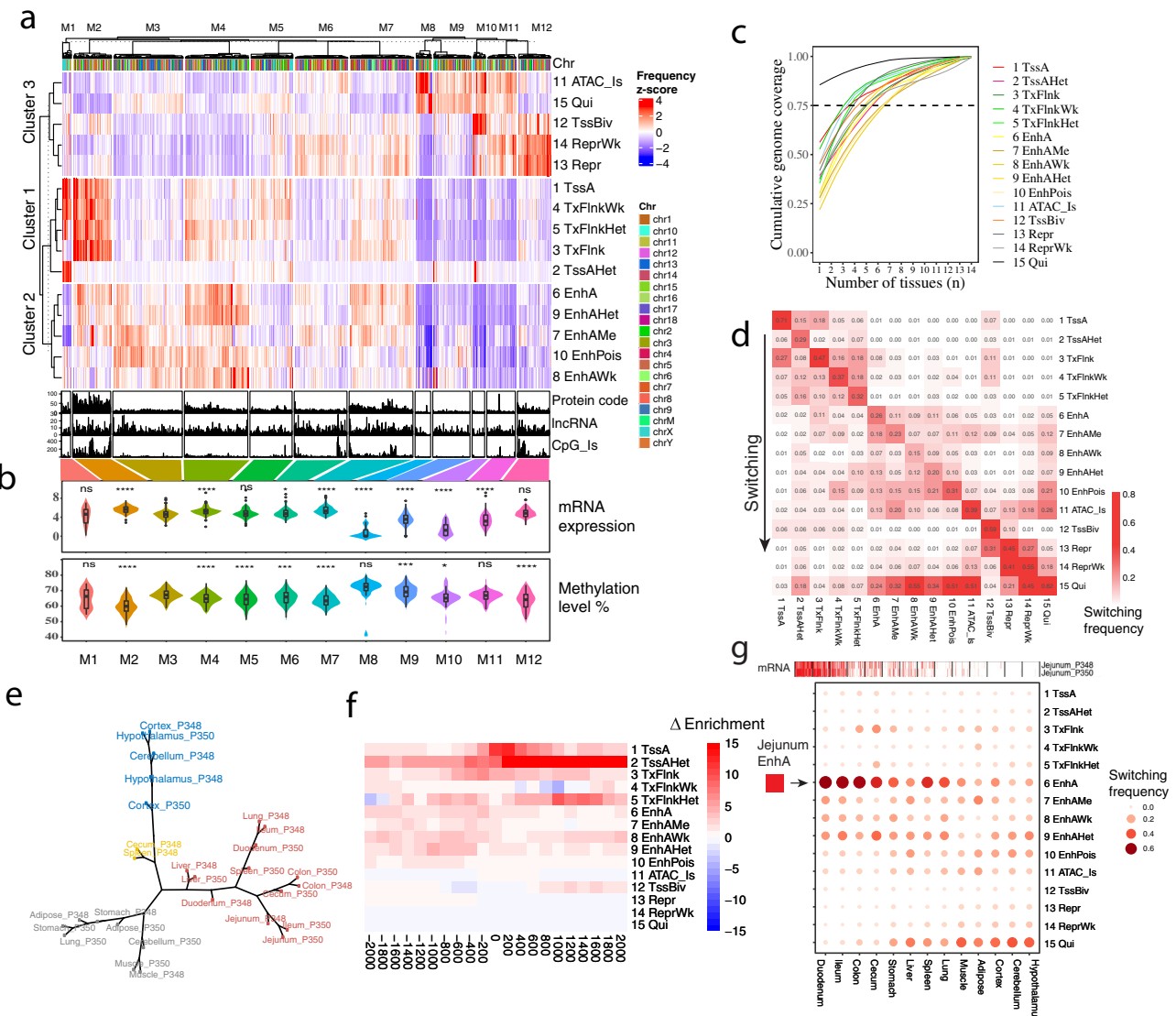

**Fig. 3 Genome-wide chromatin state dynamics across tissues. a** Clustering of 2 Mb intervals (1224 columns) into modules (M1–M12) based on average chromatin state frequency across tissues in each interval. Number of protein-coding genes, lncRNA, and CpG islands in each interval shown in bottom. **b** Average mRNA expression ($\log_2(\text{TPM} + 1)$) of genes and average methylation level of 2 Mb intervals belonging to each module. M1–M12 module was comprised of 24, 100, 183, 167, 111, 139, 168, 41, 98, 33, 75, 85 intervals, respectively. M3 was used as reference for the statistical two-sided $t$-test, where *$P < 0.05$, **$P < 0.01$, and ***$P < 0.001$. $P$ values of gene expression (M1 = 0.33, M2 < 2.2e-16, M4 = 4.8e-10, M5 = 0.15, M6 = 0.017, M7 = 1e-14, M8 < 2.2e-16, M9 = 3.3e-10, M10 = 2.5e-15, M11 = 8.6e-08, M12 = 0.08); $P$ values of methylation level (M1 = 0.066, M2 < 2.2e-16, M4 = 6.7e-09, M5 = 8.1e-07, M6 = 0.00027, M7 < 2.2e-16, M8 = 0.1, M9 = 0.00028, M10 = 0.049, M11 = 0.26, M12 = 5.5e-07). No adjustment was made for multiple comparisons. Whiskers show 1.5× interquartile range. Black circles were outliers. **c** Chromatin state variability based on cumulative genome coverage fraction. Dashed line = 0.75. **d** Chromatin state switching between all tissues. **e** Hierarchical epigenome clustering using H3K4me1 signal in EnhA states. **f** Chromatin state enrichment in promoters of genes with jejunum-specific expression, relative to muscle. **g** Chromatin state switching of target enhancers (EnhA) of jejunum-specific genes in other tissues.

for histone modification Gene Ontology (GO) terms (Supplementary Data 3). This may indicate potential roles of TssAHet in heterochromatin on the X chromosome[37]. In addition, we observed that M3 and M12 had similar level of mRNA expression, but opposite directions in terms of enrichment for "repr" and "enh" marks. However, M12 had a significantly lower methylation level than M3, indicating that methylation may play an independent role in gene regulation.

By examining the distribution of chromatin states among all 14 tissues, we found that enhancer activity was the most variable between tissues, whereas promoter activity was least variable (Fig. 3c, d and Supplementary Fig. 5d,e). Among promoters, TssBiv was least constitutive and often switched to TSS-proximal

transcribed or quiescent regions between tissues (Fig. 3d). Hierarchical clustering of samples using the signal intensity of H3K4me1 within EnhA clearly separated different tissue types (Fig. 3e), as well as H3K27ac in EnhA, H3K4me3 in TssA, and H3K27me3 in Repr (Supplementary Fig. 6), suggesting that the signal intensity of individual epigenetic marks in corresponding regulators is highly indicative of tissue identity.

To explore the relationship between proximal regulatory elements (within 2 kb of TSS of genes) and tissue-specific gene expression, we identified genes with tissue-specific expression (TSE), which were significantly engaged in known biological functions of specific tissues (Supplementary Fig. 7 and Supplementary Data 4). We also observed that TSE genes were enriched

for active states (promoters, transcribed regions, and enhancers) and depleted for repressed states in the 2 kb regions around their TSS in the corresponding tissue compared to other tissues (Fig. 3f). Furthermore, we found that predicted target enhancers (Supplementary Data 5) of TSE genes were more constitutive among biologically similar tissues compared with other tissues (Fig. 3g), which was consistent with promoters of TSE genes (Supplementary Fig. 8).

**Functional characterization of tissue-specific chromatin states.** As enhancers were most variable among tissues compared to other chromatin states, we identified an average of 6895 tissue-specific EnhAs among 14 tissues, ranging from 1393 in jejunum to 14,811 in skeletal muscle (Fig. 4a). To further investigate the biological functions of such enhancers, we defined three other types of EnhA, including all-common EnhA (shared among all tissues), gut-common EnhA (shared among gut tissues), and brain-common EnhA (shared among brain tissues). Gene Ontology (GO) analysis of putative target genes of these different types of EnhAs revealed distinct biological functions (Fig. 4b and Supplementary Data 6). For instance, all-common EnhAs were involved in fundamental biological processes (e.g., regulation of mRNA catabolic processes and wound responses), whereas gut-common EnhAs were significantly involved in intestinal development, digestion and absorption, and immune response. EnhAs that were specifically active in individual gut tissues showed distinct functions, clearly matching the known biological functions of the tissue in question. For example, jejunum-specific EnhAs were involved in biological processes relevant to T cell and lymphocyte function[38], whereas colon-specific EnhAs were mainly engaged in stress-activated MAPK cascades[39] (Fig. 4b). We observed that intestine- and spleen-specific EnhAs shared many immune functions, and brain-specific EnhAs were significantly involved in memory and learning (Fig. 4b). Furthermore, we observed that putative target genes of tissue-specific EnhAs were specifically highly expressed in the corresponding tissues (Fig. 4c), and methylation levels of tissue-specific EnhAs were lower in the corresponding tissues (Supplementary Fig. 9a). These results suggest that the activity of these tissue-specific enhancers and their methylation level accurately predicted the expression of associated target genes.

To explore potential tissue-specific transcription factors (TF), first we identified motifs that were significantly enriched in tissue-specific EnhAs (Fig. 4d and Supplementary Fig. 10a), such as that of MEF2A and SIX1 in muscle, SOX10 in brain, and HNF1B and HNF4A1 in liver and intestinal tissues (Supplementary Fig. 11), all of which was consistent with previous findings in humans[8]. In addition, we found the binding motif of HNF4G, which participates in the renewal of intestinal stem cells in mice[36] and is specifically active in intestine (Supplementary Fig. 4b), was enriched in intestine-specific EnhAs. Moreover, CDX2, a major regulator of intestine-specific genes involved in cell growth and differentiation, is highly expressed in jejunum compared to duodenum and ileum[40,41], and its motif is specifically enriched in jejunum-specific EnhAs. The expression levels of the inferred TFs were higher in the corresponding tissue than in other tissues (Supplementary Fig. 10b, c), indicating that these tissue-specific enhancers are hotspots for TF activity and play important roles in the tissue-specific regulation of gene expression. We further observed that genes linked to tissue-specific EnhAs were significantly associated with biologically relevant complex diseases in humans and mice (Fig. 4e, Supplementary Fig. 9b, and Supplementary Data 7). For example, colon-specific EnhAs were associated with diseases involving recurrent bacterial

infections, and cecum-specific EnhAs were significantly associated with diseases involving bruising susceptibility.

As promoters also play an important role in tissue-specific function, we also explored potential function for tissue-specific promoters (TssA) and found that promoters also showed tissue-specific regulatory (TSR) function, but to a lesser degree than enhancers (Supplementary Fig. 12 and Supplementary Data 8).

**Chromatin state predictions enhance the biological interpretations of adaptive evolution and complex traits in pigs.** To determine whether genomic regions associated with adaptive evolution are significantly enriched in regulatory elements (REs), we first identified 11,329 selection signatures (the top 5% of regions measured by $F_{ST}$) by comparing wild with domesticated pigs in Asian and European populations across 406 whole-genome sequencing datasets (Supplementary Data 9, 10, and 11). We found that genomic regions under selective pressure were most enriched for TssA and TSS-proximal transcribed regions, followed by enhancers, with similar patterns in both Asian and European populations (Fig. 5a, Supplementary Fig. 13a). In examining tissue-specific regulation, our analysis revealed that the all-common TssA were significantly enriched within regions under selective pressure in both populations (Fig. 5b). Interestingly, spleen-specific REs were most enriched in Asian pig domestication, whereas cortex-specific REs were most enriched in European pig domestication (Fig. 5b). Consequently, tissue-specific gene regulation may have played an essential role in the adaptive selection processes that resulted in Asian and European pig domestication. This result was also in agreement with the general observation that Asian domesticated pigs could be more resistant to malaria[42,43], whereas European domesticated pigs are more active and aggressive[44,45].

To ask whether SNPs associated with complex traits in pigs are enriched in regulatory regions, we integrated GWAS signal enrichment analysis for 44 complex traits (Supplementary Data 12) with all 15 chromatin states, and demonstrated that GWAS signals were most enriched in TssA (Fig. 5c), which was consistent with previous findings in humans[46]. We also found that enrichment for variants associated with complex traits was significantly positively correlated with signatures of selection (Supplementary Fig. 13b, c). We then asked if tissue-specific REs were involved in genetic control of specific complex traits. To answer this question, we conducted GWAS signal enrichment analysis for average daily gain (ADG) in three separate breeds (i.e., Duroc, Landrace, and Yorkshire), with emphasis on tissue-specific TssA and EnhA. As we expected, muscle, adipose, liver, and gut-common regulatory elements were the most relevant for ADG (Fig. 5d). In further examining the top ADG QTLs in Landrace (Fig. 5e), we identified a top hit SNP located in a muscle-specific EnhA (Fig. 5f). Based on CTCF loop target gene prediction and Hi-C loop interaction, this EnhA potentially targets *ZNF532* and *ALPK2*. Among all seven genes within this QTL, *ALPK2* plays important roles in cardiogenesis and was upregulated in the *longissimus dorsi* muscle in Wannanhua compared with Yorkshire pigs[47,48], and was the only gene specifically expressed in muscle (Fig. 5f–h). These results suggest the SNP in this muscle-specific EnhA may regulate *ALPK2* expression and is a candidate causal variant contributing to ADG. Additional evidence from eQTLs in muscle showed the highest enrichment in accessible enhancers (EnhA and EnhAMe) compared to other chromatin states (Supplementary Fig. 13d), suggesting that genetic regulatory variants are more likely to influence gene expression by perturbing enhancers. In summary, these results together demonstrate the important role of

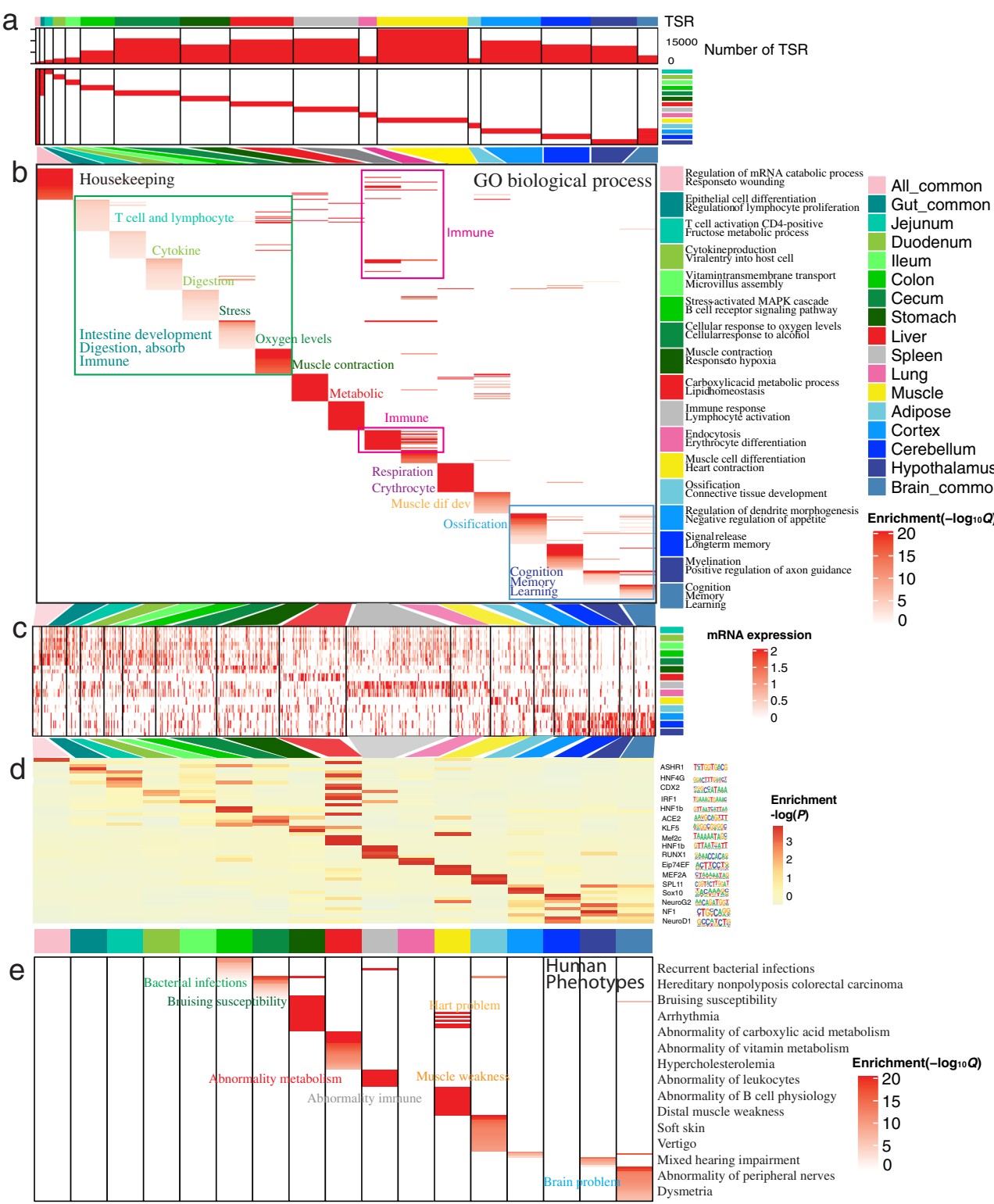

functional genome annotation for interpreting the molecular mechanisms underpinning complex traits, adaptive evolution, and gene regulation.

**Comparative analysis of pig, mouse, and human epigenomes.** The distribution of individual epigenetic marks and chromatin accessibility with respect to genomic features (e.g., 5'UTRs and exons) was consistent between pig, mouse, and human (Supplementary Fig. 14). To determine if chromatin states are similarly conserved between these species, we predicted 15 chromatin states in mouse and human based on the same epigenetic marks in pig. The resulting chromatin state predications demonstrated general similarity in chromatin states among the three species in terms of genome coverage, genomic distribution, and sequence conservation (Fig. 6a and Supplementary Fig. 15).

To explore the relationship between the epigenome and DNA sequence conservation among these three species, we divided each genome into regions corresponding to 50 different levels of

**Fig. 4 Tissue-specific strong enhancers (EnhA) and their potential functions in 14 tissues. a** The number and enrichment distribution of 17 modules of TSR (strong enhancers (EnhA)) in tissues. TSR tissue-specific regulatory elements. The top colors represent 17 modules of strong enhancers (column) referred to by the legend on the right. The side colors represent 14 tissues (row), also referred to by the legend on the right. **b** Functional enrichment of proximal genes for each module based on gene ontology (GO) biological processes. The columns represent 17 modules of strong enhancers. The rows represent GO terms in each module. All GO terms are presented in Supplementary Data 5. Notes within the heatmap summarize functions of nearby GO terms (up-noted from jejunum to spleen, down-noted for lung, muscle, and adipose). **c** The average expression (TPM) of EnhAs' putative target genes in each module. The columns represent the genes in each module, the rows represent each tissue. **d** The enrichment of transcription factor motifs in each module. The columns represent 17 modules of EnhAs. The rows represent motifs. All enriched motifs are presented in Supplementary Fig. 10a. The P values were generated by HOMER. **e** Enrichment for human phenotypes in each module, based on proximal genes. The columns represent 17 modules of EnhAs. The rows represent phenotypes. The enrichment of all phenotypes is presented in Supplementary Data 7. Notes within the heatmap summarize nearby enriched phenotypes, with the color of the text indicating the corresponding tissue.

sequence conservation (0–49th) (Methods section). Our results revealed that the majority of chromatin states showed higher conservation levels in sequences under either rapid or slow evolution than those under neutral evolution, following a U-shaped distribution[49] (Fig. 6b and Supplementary Fig. 16a). However, some subtle differences among chromatin states were observed. For example, TssAHet, TxFlnkHet, and EnhAHet were found in the right half of the U curve, whereas TxFlnkWk and EnhAWK were in the flat bottom of the U curve (Supplementary Fig. 16b, c). Overall, we found that the densities of chromatin states and gene elements followed a similar U-shaped distribution (Supplementary Fig. 16b, c), supporting the hypothesis that conserved epi-modifications may buffer negative selective pressures by providing the genome more elastic room to adapt[49]. Furthermore, we categorized orthologous genes into 50 groups based on the degree of conservation of gene expression between species and observed that genes with more conserved expression levels also demonstrated more conserved TssA and TssBiv signatures (Fig. 6c). In further examining the sequence of extremely conserved (49th) or extremely variable regions (0th), we found that genes linked to TssA shared by human and pig were involved in basic biological processes, such as ncRNA metabolic process and mRNA catabolic process (Supplementary Fig. 17). Among extremely conserved regions (49th) in the brain, genes with human-specific TssA (e.g., *FOXG1*[50]) were engaged in neuron fate commitment, cerebral cortex development, learning and memory (Fig. 6d, Supplementary Fig. 18, and Supplementary Data 15).

Next, we evaluated the evolutionary basis of complex traits in humans. Heritability enrichment analysis of 47 complex traits across 15 chromatin states that were mapped from pigs to orthologous regions in humans found that promoters and TSS-proximal transcribed regions were most enriched for variants (Fig. 6e). We further revealed that the more conserved (species-shared) chromatin states showed significantly higher enrichment for complex trait heritability than the more divergent (species-specific) chromatin states (Fig. 6f). Then we further examined the role of tissue-specific gene regulation in human complex traits. Our heritability enrichment analysis of complex traits, based on human orthologous regions of tissue-specific EnhAs identified in pigs, demonstrated that tissue-specific enhancers were significantly enriched for the corresponding human complex traits relevant to biological functions of specific tissues (Fig. 6g). For instance, lung-specific EnhAs were significantly enriched for the heritability of lung forced expiratory volume 1 (FEV1), liver-specific EnhAs for fasting glucose and cholesterol, colon-specific EnhAs for Crohn's disease, and cortex-specific EnhAs for intelligence (Fig. 6g).

Finally, we sought to determine if this annotation of regulatory elements substantiated the use of pig as an appropriate animal model for different human diseases by comparing human, mouse, and pig epigenomes in specific tissues. In brain cortex, the mouse-

human shared EnhAs exhibited significantly higher heritability enrichment than the pig-human shared EnhAs for most brain-relevant traits, such as attention deficit hyperactivity disorder (ADHD), intelligence, depression, and reaction time, with the exception of Alzheimer's disease, for which heritability was significantly enriched in pig-human shared EnhAs rather than the mouse-human shared EnhAs (Fig. 6h). This was in line with the use of pigs as a biomedical model for studying Alzheimer's disease[21,51]. Similar observations were found in intestine (Crohn's disease and inflammatory bowel disease (IBD), but not colorectal cancer, which demonstrated more heritability in the pig-human shared EnhAs) (Fig. 6i) and in adipose (body mass index (BMI), body fat percentage, waist-hip ratio, and weight had significantly higher heritability enrichments in the pig-human shared EnhAs) (Fig. 6j). Similar results were also noted in a comparative promoter analysis (TssA) (Supplementary Fig. 19). Our findings suggest that for certain human traits, the pig could be a better biomedical model than the mouse, and vice versa.

## Discussion

In this study, we provided the most comprehensive catalog of porcine regulatory elements to date, spanning 14 tissues, including six gut-associated tissues, and characterized the dynamic chromatin state landscape across these tissues, thereby uncovering extensive tissue-specific regulation of gene expression.

The annotation of functional elements in human and mouse has proven highly effective for the identification of causative variants of complex traits[23,27], and positional candidate genes for complex traits such as feed efficiency and growth are functionally conserved across vertebrate species[52]. Our results demonstrated that variants of complex traits and eQTLs of growth-related traits were significantly enriched in the active promoters and enhancers annotated by this study. Specifically, we speculate that a potential causative SNP, which was associated with average daily gain and which was found within a muscle-specific enhancer, may regulate the expression of *ALPK2*[47,48], a gene demonstrating muscle-specific expression (0.5 Mb away). In addition, our annotation of functional elements in pigs allows us to evaluate the potential role of regulatory elements on pig domestication. Our analysis illustrated that signatures of domestication were significantly enriched in porcine regulatory elements. Specifically, genetic variants in the spleen-specific promoters were enriched during Asian pig domestication, whereas variants within cortex-specific promoters were enriched during European pig domestication. This insight may reflect the observed distinct phenotypic difference between Asian (more resistant to malaria[42,43]) and European domesticated pigs (more active and aggressive[44,45]). Further investigation is warranted to deepen our understanding of genetic selection and domestication in the pig. This regulatory element atlas will serve as a valuable source for the livestock community to inform GWAS and eQTL findings, genomic selection programs, and genome editing strategies, as well as to enhance our

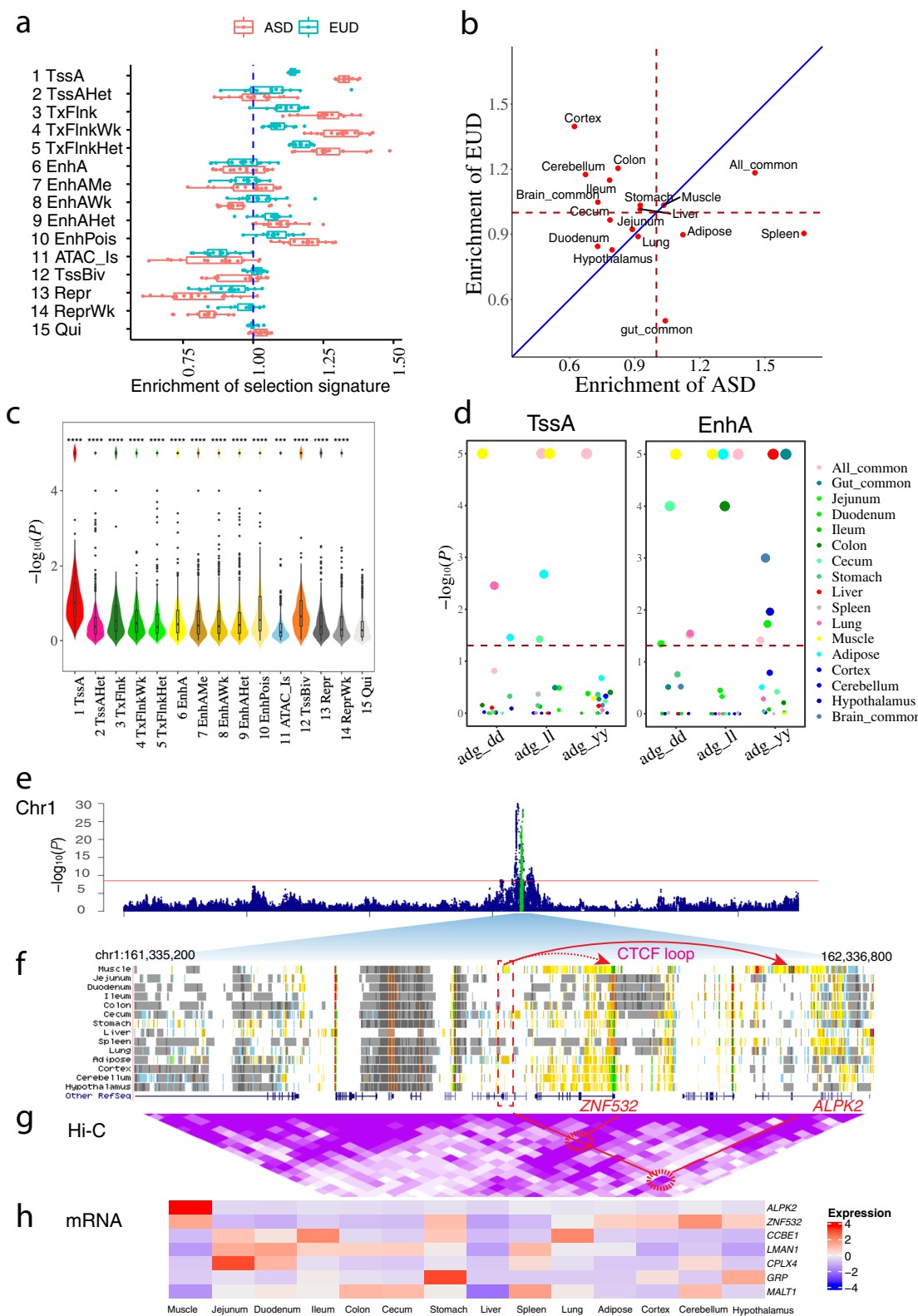

understanding of genome evolution and adaptation. With continued efforts by the FAANG Consortium[53], more epigenomic data will be available from diverse samples, such as reproductive tissues, additional developmental stages, and different physiological states. The systematic integration of "omics" data, such as the on-going pig GTEx effort, will contribute additional insight into the biological mechanisms that underpin agronomic traits,

and thereby enhancing genetic improvement of economically important phenotypes[53].

Finally, this atlas of functional elements provided a unique opportunity for comparative epigenomic analysis between human, mouse and pig, the results of which can inform which species constitute the most appropriate biomedical model(s) for specific human diseases. We observed that regions under positive

**Fig. 5 Chromatin state plays an important role in pig domestication and complex traits. a** Domestication selection signature enrichment within chromatin states in Asian and European pigs. ASD Asian pig domestication, EUD European pig domestication. Values greater than 1 (dashed line) indicate significant enrichment. Whiskers show 1.5× interquartile range. Each datapoint represents one of 14 different tissues. **b** Domestication selection signature enrichment in tissue-specific promoters (TssA) between Asian and European pigs. Values >1 (dashed line) indicate significant enrichment, measured by Fisher's exact test. Deviation from the diagonal line shows a tissue's enrichment tendency towards either Asian or European pigs. **c** Genome-wide association study (GWAS) signal enrichment within chromatin states across 14 tissues and 44 complex traits in pigs. The statistical significance of comparisons were calculated by two-sided $t$-test using "15 Qui" as a reference. No adjustment was made for multiple comparisons. ***$P < 0.001$. The $P$-value in each group were "1 TssA"<2.2e-16, "2 TssAHet"=9.1e-09, "3 TxFlnk"< 2.2e-16, "4 TxFlnkWk"=6.7e-16, "5 TxFlnkHet"=2.8e-12, "6 EnhA"<2.2e-16, "7 EnhAMe"=3.6e-16, "8 EnhAWk"=2.5e-16, "9 EnhAHet"<2.2e-16, "10 EnhPois"<2.2e-16, "11 ATAC_Is"= 0.00015, "12 TssBiv"<2.2e-16, "13 Repr"=7.1e − 15, and "14 ReprWk"=3.8e-10. Whiskers show 1.5× interquartile range. Black points were outliers. **d** GWAS signal enrichment of promoter (TssA) and strong enhancer (EnhA) tissue-specific regulatory elements (TSR) for average daily gain (ADG) in three pig populations (dd: Duroc, ll: Landrace, yy: Yorkshire). Significance was based on 10,000 iterations of a genotype cyclical permutation test. Dashed line set at −$\log_{10}(P = 0.05)$. Values over the dashed line were significantly enriched. **e** Manhattan plot of ADG in the Landrace population (88,984). **f** Chromatin states for each tissue in a genomic region where GWAS hits were found. Dashed rectangular box includes a muscle-specific enhancer that coincides with GWAS hits. Arrows in red indicate predicted CTCF looping and H3K27ac signal, which together suggest that the muscle-specific enhancer may target *ZNF532* and *ALPK2*. **g** Hi-C loop (25 kb resolution) depiction between a muscle-specific enhancer and putative target genes. Purple shading for the Hi-C data represents loop intensity (auto-scale). Two highlighted Hi-C loops delineated with red circles are potential contacts between a muscle-specific enhancer and *ZNF532* and *ALPK2*. **h** Expression (normalized and centered TPM) of genes proximal to the muscle-specific enhancer.

or negative selective pressure demonstrated higher conservation of epigenetic signatures (such as TssA, TssBiv and TxFlnk) than those which are not subject to selective pressure (i.e., the selectively neutral), further confirming the hypothesis that elasticity of regulatory conservation may play an important role in the evolution of the less conserved regions (impact of negative selection pressure)[49]. Recently evolved liver enhancers (i.e., species-specific) are often associated with genes that show evidence for being under positive selection[54]. Such enhancers have been demonstrated to actively affect gene expression, although they have a smaller effect than enhancers shared across species when the comparison is controlled for number of enhancer elements acting on a given gene[55]. However, human-specific promoters in brain tissues were enriched in intelligence-related genes, which suggests a critical role for epigenomic regulation of novel biological function in humans in the most evolutionarily conserved regions. It is widely accepted that neither mouse nor pig is universally appropriate to serve as an animal model for every human disease[18,56]. Gene regulatory networks play significant roles in controlling phenotypic variance of complex traits, including most human diseases. In examining heritability enrichment of 47 complex traits in humans, our epi-conservation analysis among three species (comparing pig-human vs. mouse-human shared enhancers in different tissues) revealed insights and potential underlying molecular mechanisms as to why pig might be a more appropriate animal model for certain human diseases than mouse, and vice versa. This line of evidence is consistent with many studies of human diseases using either mouse or pig as an animal model[18]. Our study provides a basis for understanding genetic regulation of complex traits, such as human diseases, by focusing on regulatory network conservation across different mammalian species. Although the findings from our study are intriguing, experimental studies and more epigenomic data from additional tissues, cell types, and species – such as non-human primates – will be needed to extend and functionally validate the biological mechanisms that underpin complex traits and diseases[9,49].

## Methods

**Animals and tissues**. Procedures for tissue collection followed the Animal Care and Use protocol (#18464) approved by the Institutional Animal Care and Use Committee (IACUC), University of California, Davis. Five gut-associated tissues (stomach, jejunum, duodenum, ileum, and colon) were collected from two Yorkshire littermate male pigs at six months of age from Michigan State University[29]. Cecum from two female hybrid pigs (Yorkshire-Hampshire cross, five months of

age) were obtained at University of California, Davis meat laboratory. Tissues were first flash frozen in liquid nitrogen, and then stored at –80 °C until further processing.

**Library construction and sequencing**. We performed ChIP-seq (H3K4me3, H3K4me1, H3K27ac and H3K27me3) experiments on flash-frozen tissue samples using the iDeal ChIP-seq kit (Diagenode Cat.#C01010059, Denville, NJ), as previously described[29]. Briefly, 20–30 mg powdered tissue was cross-linked with 1% formaldehyde for 8 min and quenched with 100 μl of glycine for 10 min. Nuclei were obtained by centrifugation at 2000×g for 5 min and resuspended in 600 μl of iS1 buffer for incubation on ice for 30 min. Chromatin was sheared using the Bioruptor Pico between 10 and 15 cycles depending on the tissues. For immuno-precipitation experiments, ~1–1.5 μg of sheared chromatin was used as input with 1 μg (histone modifications) or 1.5 μg (CTCF) of antibody following the protocol from the kit. The following antibodies used were from Diagenode: H3K4me3 (comes with Diagenode iDeal Histone kit), H3K27me3 (#C15410069), H3K27ac (#C15410174), H3K4me1 (#C15410037), and CTCF (#15410210). An input (no antibody) was performed for each sample. Libraries were constructed using the NEBNext Ultra DNA library prep kit (New England Biolabs #E7645L, Ipswich, MA). Libraries were sequenced on the Illumina HiSeq 4000 platform, generating 50 bp single-end reads. ATAC-seq libraries were generated from frozen tissue samples by a modified protocol (https://figshare.com/articles/dataset/Final_ATAC_protocol_docx/13891268) according to the protocol of Omni-ATAC[57] and cryopreserved nuclei[58]. The sequencing was performed on Illumina's NextSeq platform, generating 40bp paired-end reads. For the RRBS-seq experiments, DNeasy Blood & Tissue kit (Qiagen, Hilden, Germany) was used for extraction of DNA from frozen tissues. The samples were sent to Novogene (Sacramento, CA, USA) for library construction and sequencing on the Illumina HiSeq 4000 platform, generating 150bp paired-end reads. Total RNA was isolated from flash-frozen tissue by Zymo Quick-RNA™ Miniprep kit (Irvine, CA, USA). RNA-seq libraries were constructed using the NEBNext Poly(A) mRNA Magnetic Isolation Module kit (NEB #E7490) and NEBNext Ultra™ Directional RNA Library Prep kit for Illumina (NEB #E7720, New England Biolabs (NEB), Ipswich, MA) and sequenced on the Illumina HiSeq 4000 platform, generating 100 bp paired-end reads.

**Data processing and data summary**. In total, 95 new datasets, including ChIP-seq (H3K4me3, H3K4me1, H3K27ac, H3K27me3, input control), ATAC-seq, RRBS, and RNA-seq for two biological replicates of six gut-associated tissues, were generated. We also integrated the additional 144 existing pig epigenomic datasets, including ChIP-seq (H3K4me3, H3K4me1, H3K27ac, H3K27me3, CTCF, input control), ATAC-seq, RRBS, and RNA-seq in the same two male biological replicates of eight tissues (adipose, cerebellum, brain cortex, hypothalamus, liver, lung, muscle, and spleen) from our FAANG pilot project (PRJEB14330)[29], and four Hi-C pig liver datasets from a publicly available dataset (PRJEB27364)[30]. The UC Davis FAANG Functional Annotation Pipeline (https://github.com/kernco/functional-annotation) was applied to process the ChIP-seq, ATAC-seq, and RNA-seq data, as previously described[29]. Briefly, the susScr11 genome assembly and Ensembl genome annotation (v100) were used as references for pig. Sequencing reads were trimmed with Trim Galore![59] (v.0.6.5), and aligned with either STAR[60] (v.2.5.4a) or BWA[61] (v0.7.17) to the respective genome assemblies. Alignments with MAPQ scores <30 were filtered using Samtools[62] (v.1.9). For RNA-seq, gene counts were determined using htseq-count[63] (v.0.13.5), and then trimmed mean of $M$-values

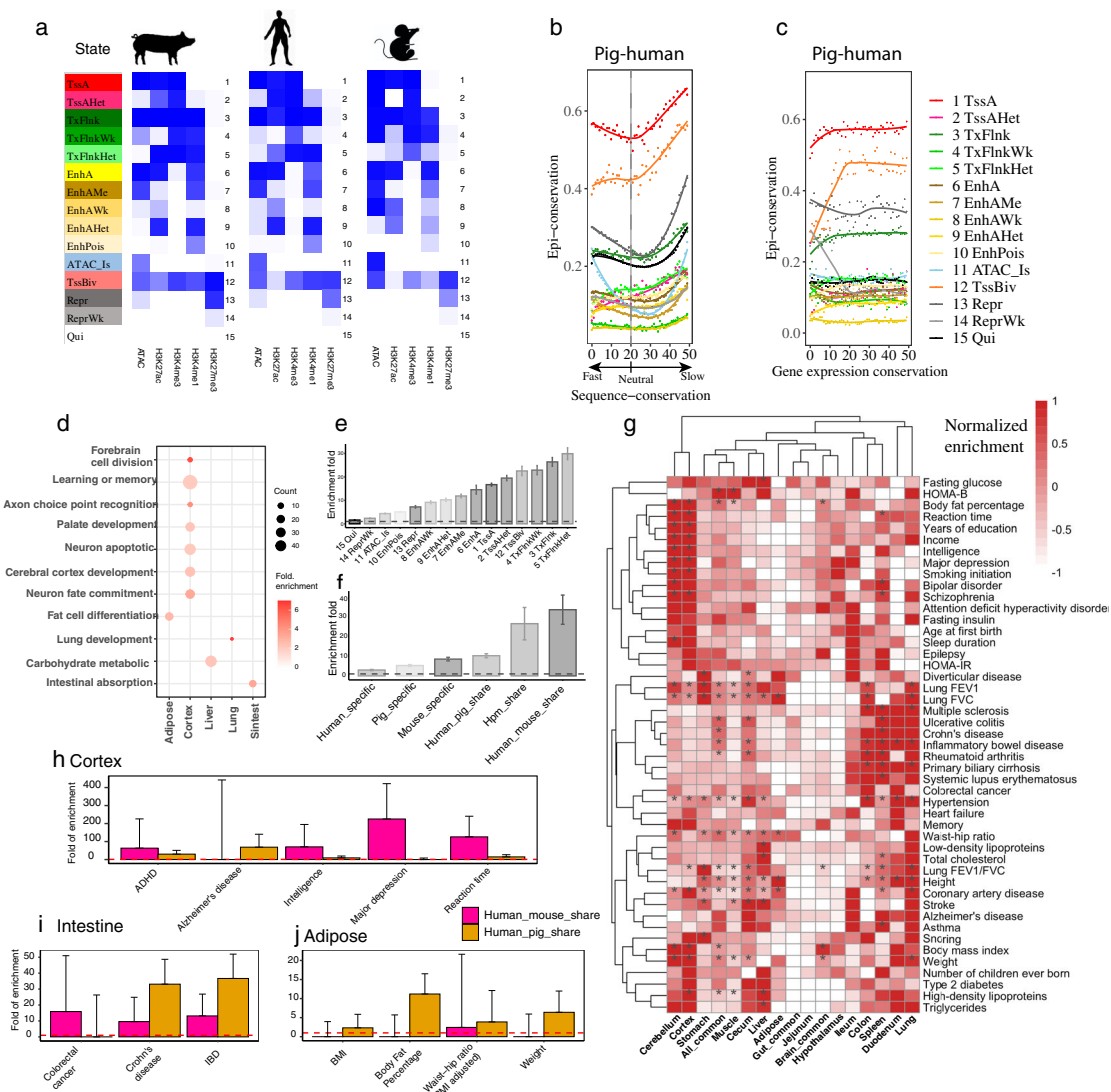

**Fig. 6 Interspecies conservation of chromatin states. a** 15 chromatin states predicated in three species. The colors from white to deep blue indicate emission probabilities, ranging from 0 to 1. **b** Relation between sequence conservation and epigenomic conservation across six tissues. Fifty genomic regions were ordered from the fastest changing (0th), neutral (20th), and slowest changing (49th) in terms of sequence conservation (Supplementary Fig. 16d). Epigenome conservation (see Methods section) for each chromatin state within each region was calculated between pigs and humans and plotted. **c** Relation between expression conservation and epigenomic conservation across six tissues. Expression conservation was based on expression of 14,302 orthologous genes among the three species. Regions were ordered from the biggest difference in expression (0th), to the smallest difference (49th). **d** GO enrichment was based on genes proximal to (±2 kb) human-specific TssA with extreme sequence conservation (49th). Count refers to the number of genes. **e** Human GWAS (47 traits) signal enrichment in 15 different chromatin states across six tissues. Enrichment was the proportion of heritability divided by the proportion of SNPs in each chromatin state. Values greater than the dashed line (set at 1) indicate significant enrichment. Error bars represent standard error around the estimates of enrichment. Dashed lines and error bars are similarly formatted in sub-figures (**f**, **h–j**). **f** Human GWAS (47 traits) enrichment in six groups of species-specific or shared EnhA across six tissues. (hpm_share stands for human-pig-mouse shared). **g** GWAS enrichment of pig tissue-specific enhancer (EnhA) in humans. "*" indicates FDR < 0.05. **h–j** Different GWAS enrichments between human-pig and human-mouse shared strong enhancers (EnhA) in brain cortex (1799 vs. 61 enhancers), small intestine (5311 vs. 2430 enhancers), and adipose (2014 vs. 1638 enhancers), respectively. Data in Figs. **e–j** are available at https://doi.org/10.6084/m9.figshare.16531197.v1.

(TMM) and transcript per million (TPM) normalization were performed using EdgeR (v3.32.0) and StringTie2 (v.1.3.3), respectively[64]. For ChIP-seq, after the filtering, duplicates were marked and removed using Picard (v.2.18.7). Regions of signal enrichment ("peaks") were called by MACS2[65] (v.2.1.1). Various quality metrics (e.g., Jensen-Shannon divergence (JSD), Supplementary Data 1) were calculated following the method described by Kern et al.[29]. RRBS data were processed using Bismark[66] (v.0.22.1) with parameters set in the RRBS pipeline (https://github.com/zhypan/Functional-Annotation-of-Pig). Hi-C contacts were called using the Juicer pipeline[67] with default parameters.

The global correlations among assays, tissues, and biological replicates were performed by deepTools[68] (v.3.5.0). Briefly, the Z-score normalized read signals of all samples within 1 kb windows were calculated by multiBigwigSummary and were presented by plotCorrelation. The signal of marks along with protein-coding genes were generated by deepTools[68] (computeMatrix scale-regions function) with parameters -a 2500 -b 2500. The Z-score was used to normalize bigWig of five marks given input files.

**Annotation of chromatin states**. ChromHMM[69] (v.1.20) was used to train the chromatin state prediction model by integrating ChIP-seq (H3K4me3, H3K4me1, H3K27ac, H3K27me3, and input control) and ATAC-seq data from two biological replicates of 14 tissues. The same tissue of two biological replicates were collectively considered as one tissue epigenome. The 15-state model was chosen, as it presented maximum number of states with distinct epigenetic mark combinations. We

labeled these 15 chromatin states based on their combinations of histone modifications and enrichment around TSS[8,27]. Then the fold enrichment of each chromatin state for each external gene element (e.g., exon, CpG islands) was calculated by (C/A)/(B/D), where A, B, C, D are the number of bases in a chromatin state, a gene element, overlapped between a chromatin state and a gene element, in the genome, respectively. In addition, we also computed chromatin state fold enrichment in mammalian conserved elements which identified from Multiple Sequence Alignments (MSA) using the Genomic Evolutionary Rate Profiling (GERP) software based on 103 mammals (https://ftp.ensembl.org/pub/release-100/bed/ensembl-compara/103_mammals.gerp_constrained_element/). The methylation level of each state and its up- and down- stream 10 kb region was calculated by the computeMatrix scale-regions function of deepTools with parameters–binSize 500,–regionBodyLength 2000 and–skipZeros.

**Clustering of large-scale chromatin structure.** To examine genome-wide chromatin structure, we first divided the genome (excluding contigs) into 1224 fragments of 2 Mb in length, following the Roadmap Epigenomics project analysis[8]. Then we calculated the state frequencies (state bin/total bin) in each 2 Mb fragment for each tissue, and then the average frequency across tissues. To identify modules, column clustering was performed by k-means = 12, and rows were clustered using $k = 3$. In addition, we calculated number of protein-coding genes, lncRNA, and CpG islands for each 2 Mb fragment by BEDTools[70] (v.2.29.2). We also calculated the average TPM and average methylation level of protein-coding genes across 14 tissues in each 2 Mb fragment. Then the average gene expression and methylation level in each of the 12 modules were calculated and a Student's t-test was performed with parameter setting ref.group = "M3". M3 was used as a reference, since it was closest to median expression.

**Chromatin state variability.** For each state, we first obtained regulatory regions across 14 tissues (RRATs) (Supplementary Fig. 4a–c) using the BEDtools merge function (any regulatory region between two tissues overlapped by 1 bp was merged), then we calculated the total genomic length for each tissue (GL) and the total combined genomic length (TGL) for RRATs. The relative state coverage per tissue was derived by GL/TGL (Supplementary Fig. 5d). Finally, by following the order from high to low based on the GL/TGL value in each tissue, we calculated the total genomic length of accumulated tissues (aGL) by adding one tissue at a time until all 14 tissues were added, and the cumulative state coverage was calculated as aGL/TGL. States whose cumulative coverage changed faster than others were considered to be less constitutive (more variable) states.

**Chromatin state switching between tissues.** Chromatin state switching between tissues was calculated by pairing two tissues. Given a pairing of tissues A and B, we first counted total bins of chromatin state "e" in A (TbAe), then obtained the overlap bins of chromatin state "e" (Obe) in A and B, then computed the state switching probabilities using Obe/TbAe for the tissue A to B transition and Obe/TbBe for the tissue B to A transition. By averaging these calculations for a pair of tissues, we obtained the pair switching probabilities. We calculated the state switching probabilities in between intestinal tissues, between brain tissues (Supplementary Fig. 8a, b), and between eight distinguishable tissues (jejunum, brain cortex, adipose, liver, lung, muscle, and spleen).

**Hierarchical epigenome clustering.** We first calculated an epigenetic mark's signal confidence scores ($-\log_{10}$(Poisson P value)) within 200 bp of the genomic regions for each mark of each sample as described in http://jvanheld.github.io/stats_avec_RStudio_EBA/practicals/02_peak-calling/peak-calling_report.html#data_sets. Then, we extracted a specific mark's signal confidence score of each sample for specific state of RRATs regions. For example, we extracted H3K4me1 signal confidence scores for EnhA. After combining all samples' mark confidence scores for each tissue and each state, we constructed a distance matrix using the Ward D2 linkage method with hierarchical clustering and Euclidean distance in R.

**Promoter enrichment analysis of tissue-specific expressed genes among 14 tissues.** To evaluate how chromatin state changes at promoter regions of TSE genes across tissues, we performed a Student's t test among 14 tissues to identify tissue-specific expressed genes, using the same method described by Fang et al.[71] and Finucane et al.[72]. We first grouped some tissues into different sub-groups, such as small intestine (jejunum, ileum, and duodenum), large intestine (cecum, colon), and brain (cortex, cerebellum, and hypothalamus). Then we scaled the log2-transformed expression (i.e., log2TPM) of genes to have a mean of zero and variance of one within each tissue group. Further we computed a t-statistic to identify tissue-specific expressed genes by excluding the tissues in the same sub-group. Last we selected genes with the top 5% t-value as TSE genes[71]. Several other methods could be also used to detected tissue-specific genes[73]. Enrichment of GO biological process terms for these TSE genes was conducted by WebGestalt 2019[74] (http://www.webgestalt.org/) using the default significance level (FDR < 0.05). Then we calculated the chromatin state fold enrichment of TSE genes (2 kb region up-

and down-stream of TSS) in each tissue, and the change in TSE enrichment in a given specific tissue minus other tissues.

**Chromatin state switching of target enhancer (EnhA) of TSE genes.** To evaluate how enhancers of TSE genes switch among tissues, we first identified the target enhancers of TSE genes following the method described in our recent study[29]. Briefly, we first generated CTCF-mediated loops from CTCF ChIP-seq data by FIMO[75] following the method described in Oti et al.[76]. Then the nested and overlapping CTCF loops were merged to form the predicted CTCF loops. We then predicted the enhancer-gene pairs according to the Spearman's rank correlation of every possible combination of regulatory element H3K27ac signal and gene expression value within each loop. Benjamini–Hochberg adjustment (FDR < 0.05) was used to define putative interacting pairs (Supplementary Data 5). The enhancers in the enhancer-gene pairs that corresponded to TSE genes were considered as TSE genes' target enhancers. Finally, we computed enhancer state switching probabilities of TSE genes among tissues using the method described above.

**TSR of enhancers and promoters and their putative functional regulation.** For strong enhancers (EnhA) identified in each tissue, we counted the bins of overlapping RRATs by comparing to other tissues. If the number of bins > = 1 in a given tissue and a given RRAT, the RRAT region would be assigned a value of 1 for this tissue; otherwise it was assigned 0. We generated a total of 17 modules of tissue-specific regulatory element (TSR) enhancers. These 17 modules included all-common (presented in all tissues), gut-common (presented in all 5 intestinal tissues), brain-common (presented in all 3 brain tissues) and 14 tissue-specific modules. The same method was used to obtain TSR for promoters (1_TssA). In addition, we performed enrichment analyses (GO, Human Phenotype Ontology (HPO), Mouse Phenotype) based on genes proximal to TSR using the GREAT[77] tool with default parameters, except for TSR promoters (proximal 2 kb upstream, 1 kb downstream, plus distal up to 3 kb). We used a cut-off of FDR < 0.05 for both the binomial and the hypergeometric distribution-based tests.

The motifs of tissue-specific EnhAs were identified by HOMER[78] (v.4.11) with cutoff FDR < 0.05. We selected the top three enriched or tissue function-relevant motifs for each tissue as the candidate tissue-specific EnhAs motifs and identified a total of 51 motifs enriched in tissue-specific EnhAs. In addition, we used these 51 motifs as known TF motifs to conduct the enrichment for all tissues by HOMER. The mRNA expression of corresponding TFs in pigs were used to calculate the correlation with motif enrichment.

**Selection signature enrichment analysis of chromatin states.** A total of 406 whole genome sequence (WGS) datasets (Supplementary Data 9) in pigs (Asian wild (58) and domestic pigs (129), European wild (35) and domestic pigs (184)) were trimmed by Trimmomatic[79] (v.0.39), mapped by BWA (0.7.17), and marked for duplicates by GATK[80] (v.4.1.4.1) MarkDuplicates with default parameters. The genome variant calls for each sample were called by GATK HaplotypeCaller. All genome variant calls were then combined and the variants for each sample were called by GenotypeGVCFs. After SNP calling, the variants were filtered using VariantFiltration (QD < 2.0, MQ < 40.0, FS > 60.0, SOR > 3.0, MQRankSum < −12.5, ReadPosRankSum < −8.0) to remove low-quality SNPs. We then performed $F_{ST}$ analysis between Asian wild and domestic pigs, and between European wild and domestic pigs with a 10-kb sliding window and 10 kb step using popgenWindows.py (https://github.com/simonhmartin/genomics_general). We chose the top 5% of $F_{ST}$ regions as candidate selection signatures, and a total of 11,329 selection signatures with a size of 10 kb were identified. Last, we calculated the fold enrichment of selection signatures for chromatin states using the same method for gene element enrichment described above: (C/A)/(B/D). We calculated the significance of enrichment using Fisher's exact test.

**GWAS and eQTL signal enrichment of chromatin state.** The pig GWAS data of 44 traits was described previously[81]. Briefly, more than 100,000 pigs (Supplementary Data 12) were genotyped by a variety of Porcine chip arrays (8.5, 43, 60, and 70 K). And then these genotyped animals were imputed to genome-wide level using an intermediated reference panel of 474 animals genotyped by a 658K SNP array, then a reference panel of 217 WGS datasets. Furthermore, we filtered out all SNPs with either (1) a minor allele frequency below 0.5%, (2) a large deviation from Hardy–Weinberg proportions ($P < 1.0^{-6}$), or (3) an R2 value of the imputation accuracy estimated by Minimac4 less than 0.4. Last, we performed GWAS signal enrichment of 44 pig complex traits (3 ADG-related, 20 lipid-related, and 21 feed efficiency-related) for each chromatin state across 14 tissues by applying a genotype cyclical permutation test, repeated 10,000 times[71]. The eQTL data in pig muscle[82] with FDR < 0.05 were used to calculate the fold enrichment for the chromatin states using the same method described above: (C/A)/(B/D).

**Interspecies conservation of chromatin state.** We collected data from ENCODE[4,9], Roadmap Epigenomics[8] and published articles[83] (9 tissues in human and 7 tissues in mouse, Supplementary Data 13 and 14), including ChIP-seq

(H3K4me3, H3K4ac, H3K4me1, H3K27me3, and Input), ATAC-seq, DNase-seq, and RNA-seq. In total, we obtained six matched tissues (small intestine, liver, spleen, lung, adipose, brain cortex) among pig, human, and mouse. All data were processed following the same pipeline used in pig. The GRCh38 (human) and GRCm38 (mouse) assemblies with Ensembl annotations (v100) were used for data analysis. Chromatin states of human and mouse were also trained by ChromHMM and 15 chromatin states were identified. To explore the relationship between sequence conservation and epi-conservation among the three mammals, we first divided the genome into 50 equally sized sets (0th-49th) with increasing average PhyloP scores using the method detailed by Xiao et al.[49]. Briefly, the human genome was divided into 15 million 200 bp segments. Then average PhyloP scores (100 vertebrate genomes[84]) were computed for each 200 bp segment. These genomic segments were divided into 50 equally sized sets from the fastest changing sequence (smallest PhyloP scores) to the most conserved (greatest PhyloP scores). (Supplementary Fig. 13d). To quantify epigenomic conservation, we downloaded whole genome alignment UCSC chain files among human (hg38), pig (SusScr11), and mouse (mm10), and then processed as described in the UCSC Genome Wiki website (http://genomewiki.ucsc.edu/index.php/HowTo:_Syntenic_Net_or_Reciprocal_Best) to derive reciprocal best chains. Then we converted genomic coordinates between assemblies using the UCSC Liftover tool (https://genome.sph.umich.edu/wiki/LiftOver) based on 0.65 sequence identity. All chromatin states in pig and mouse were lifted over to human. The conservation rate (0–1) of each region of each state from pig to human was calculated based on state region coverage of pig over human. If there was no overlap it was assigned 0, if completely occupied it was assigned 1. The same analysis was conducted for pig to mouse and mouse to human. Furthermore, we performed genomic and epigenomic conservations for every pair of mammalian species in each tissue. Finally, we conducted the same analysis on mammalian conserved score based Genomic Evolutionary Rate Profiling (GERP) using 103 mammalian genomes (https://ftp.ensembl.org/pub/release-100/compara/conservation_scores/103_mammals.gerp_conservation_score/).

To examine the biological relevance of regions with extremely variable sequence (0–2th sets) or highly conserved sequence (47–49th sets), we extracted the human-pig shared and human-specific chromatin state TssA from these sets. Then, using the GREAT tool with parameter of proximal 2 kb upstream, 1 kb downstream, plus distal up to 3 kb was used to conduct GO function enrichment analysis.

**Expression conservation versus epi-conservation**. The TPM of 14,302 orthologous genes from pig, human, and mouse were used to identify differentially expressed genes in each tissue using the Student's $t$-test. We sorted the genes by p-value within each species and divided them into 50 equally sized sets. Then we calculated the average epi-conservation score of states in the 20 kb region around the TSS of gene in each set.

**Heritability enrichment of human complex traits in chromatin state**. To explore how conserved or species-specific chromatin states affect complex traits in humans, we extracted six types of species-shared or species-specific regulatory elements (all_shared, human_mouse_shared, human_pig_shared, human_specific, mouse_specific, pig_specific). We then performed heritability enrichment analysis by applying stratified linkage disequilibrium score regression (LDSC) to partition heritability of 47 human complex traits into distinct functional categories[46]. Stratified linkage disequilibrium score regression (LDSC) is a commonly used approach to partition the heritability of functional annotations and to estimate the enrichment degree (i.e., the proportion of heritability explained by a functional annotation (e.g., the conserved enhancers) divided by the proportion of SNPs in this annotation) based on the GWAS summary statistics[46,85]. It takes the population stratification factor into account by using regression modeling to quantify the relationship between linkage disequilibrium and the test statistic ($\chi^2$ association statistic) of SNPs from GWAS, thereby improving the power of the analysis and capturing true polygenic signal. In this study, LDSC was used to determine the SNP-based heritability estimates, and then partition the heritability into separate functional categories to demonstrate the disproportionate contribution of different functional categories to the heritability of human complex traits and diseases. These functional categories included six types of species-shared and species-specific regulatory elements, chromatin states of each tissue, and TSR of EnhA and TssA. We calculated the stratified LD scores using 1000 G Phase 3 European human samples, where only HapMap3 SNPs with INFO ≥ 0.9 and MAF > 0.05 in 1000 G European samples were used (the 1000 G samples and default SNP weights were obtained from https://github.com/bulik/ldsc).

The GWAS summary statistics for 47 human complex traits were obtained from public databases (Supplementary Data 16), with an average sample size of 321,978 (all European ancestry) and a high-quality overlap with HapMap3 panel. In addition, these GWAS results had a mean $\chi^2$ statistic >1.02 and a heritability Z-score >4[85]. We also performed default quality control for GWAS summary statistics by LDSC to remove GWAS SNPs with MAF ≤ 0.01, genotype call rate ≤0.75, INFO ≤ 0.9, an out-of-bounds P-value, duplicated Rsid, strand ambiguous variants or an extremely large $\chi^2$ statistic[85]. The results of LDSC regression for the base model, which has not been partitioned for heritability, are available in Supplementary Data 17.

**Reporting summary**. Further information on research design is available in the Nature Research Reporting Summary linked to this article.

## Data availability
High-throughput sequencing data for six gut tissues generated in this study were deposited in European Nucleotide Archive (ENA) with accession number PRJEB37735. High-throughput sequencing data of eight tissues used in this study are available the Gene Expression Omnibus (GEO) under accession GSE158430. And all the RRBS datasets were deposited in European Nucleotide Archive (ENA) with accession number PRJNA762083. All raw data are also available through the FAANG portal (https://data.faang.org/dataset). All processed data are publicly available at http://farm.cse.ucdavis.edu/~zhypan/Nature_Communications_2021 and https://doi.org/10.6084/m9.figshare.13480425. Chromatin states of pig, mouse, and human are available through the UCSC Genome Browser: http://genome.ucsc.edu/s/zhypan/susScr11_15_state_14_tissues_new; http://genome.ucsc.edu/s/zhypan/mm10_7tissues_chr_state; http://genome.ucsc.edu/s/zhypan/hg38_9tissue_chr_state.

## Code availability
The pipeline for RNA-seq, ATAC-seq, DNase-seq, and ChIP-seq processing is available at GitHub[86] (https://github.com/kernco/functional-annotation). The RRBS pipeline and other processing codes are publicly available at GitHub[87] (https://github.com/zhypan/Functional-Annotation-of-Pig).

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

## Acknowledgements

This study was supported by Agriculture and Food Research Initiative Competitive Grants no. 2018-67015-27501 (C.K.T., H.Z., C.E., and P.R.) and no. 2015-67015-22940 (H.Z. and P.R.) from the USDA National Institute of Food and Agriculture, Multistate Research Project NRSP8 and NC1170 (H.Z.), and the California Agricultural Experimental Station (H.Z.).

## Author contributions

H.Z., L.F., Z.P., P.R., C.W.E., and C.K.T. conceived and designed the study. C.E., Y.W., G.C., and Z.P. were responsible for sample collection. Z.P., Y.W., and M.H. performed ChIP-seq, ATAC-seq, and RNA-seq. Z.P., N.T., and K.W. contributed to RRBS data collection. Z.P., Y.Y., L.F., and C.K. conducted bioinformatic analysis. Z.C., G.S., GS.S., M.S.L., M.F., and P.KM. were responsible for pig GWAS data analysis. H.Y. and L.B. were responsible for pig selection signature analysis. Z.P., L.F., Y.Y., and H.Z. wrote the initial draft of the manuscript. M.H., C.W.E., P.R., and C.K.T. revised the manuscript. All co-authors contributed to the final manuscript.

## Competing interests

The authors declare no competing interests.
