## [Peer Review File · Nature Communications]

Pig genome functional annotation enhances the biological interpretation of complex traits and human diseaseReviewers' Comments:

Reviewer #1:

Remarks to the Author:

I have seen this manuscript before, and this version has been significantly improved now. Generally, this work shows us a comprehensive epigenomic landscape for pig and compared the epigenomic evolution patterns among pig, mouse and human. The data resources could be used for further multi-omics research about animal evolution and livestock breeding, and scientifically shed some new insight into mammal epigenomic evolution. I'd like to recommend publication of this article after a few modifications.

1. The authors split the genome into 2 Mb fragments and clustered them into 12 modules in L176-L187. However, how was the fragment size determined? Is there any biological meaning like chromosome location in these modules? More explanation is need in this section.

2. In Supplementary Fig. 2, duplicates of some tissue cluster badly in the PCA analysis. The data of stomach show huge difference in almost all seq data between two replicates. As there are only 2 replicates for each tissue, this abnormal value would not be processed and infect the downstream analysis. It'd be better if the authors can explain this in text.

3. Line 124. In the figure 1e, it seems that the H3K4me1 is not significantly enriched in TSS. It looks like it is enriched at about 1kb upstream of TSS?

4. There are too many heatmaps in main article. The authors may find more way to visualize their data efficiently and beautifully.

5. The p value of figure 3b should have a clearer display.

6. In figure4, why highlight the "Duodenum"?

Reviewer #2:

Pan et al. provide the most comprehensive catalogue to date of regulatory elements in the pig by integrating 223 epigenomic and transcriptomic data sets, representing 14 biologically important tissues. They defined 15 distinct chromatin states by combining five epigenetic marks across 14 tissues. All the processed data have been integrated in the UCSC genome browser, allowing research scientists interested in specific genes to benefit from these data. Moreover, the authors provide novel biological insights at different levels (tissue specificity – selective signature – GWAS signal related to traits – conservation across species). Overall, this article is innovative and provide a useful resource for the genome functional annotation field.

I have some comments and suggestions that could facilitate the reading and understanding of the manuscript, the reproducibility of the analyses and clarify some figures.

*** The 4 mark and ATACseq processed data (Fig.1)**

- It would be useful to share the bed or bigwig processed data files for these different marks.
- Fig.1b,c, in addition to peak number and genome coverage barplots across the 14 tissues, could you also provide barplots of the different mark sizes (and their average in the text).
- Fig.1e: The enrichment in the upstream of TSS is not as high for H3K4me1(yellow) and H3K27me3 (grey) as for the other marks which are consistent with the different chromatin states explored in the next section. Could you specify the distance used for the statistical test: 1kb? 2.5Kb? and clarify the sentence L125: “all four active regulatory marks exhibited significant enrichments in the upstream of transcription start sites (TSS) of genes”.
- Fig.1f: the orientation of the MYO1A gene is missing as in the supp fig.3. It would be interesting to see the pattern across tissues for all the marks in the main fig1 or in the supp Fig.3.
- The supp Fig.1 summarizes the data: could you please explain in the legend the nature and origin of the CTCF (16) and the Hi-C data (4) that are not a multiple of 8? i.e. what tissues and types of animals were collected for these two assays?

Chromatin landscape across 14 tissues (Fig.2)

- Fig.2g: The scale (log10) used for x-axis that is too large to have a good idea of the location of the different marks with respect to the TSS. A suggestion: could you add the distance from TSS for the major peaks (green peaks (Tx), yellow (Enh), blue (Atac) and grey (repr)) located upstream and downstream of the TSS. Idem, what distance from TSS for the valleys on both sides of the TssA and Tss1Het?
- Fig.2i; the ‘methylation’ level is not readable and we cannot observe what you wrote “regions with higher density of genes were characterized by active chromatin states and lower methylation level.”

Genome-wide chromatin state dynamics across tissues (Fig.3)

- Fig.3a,b: Could you add a legend for the horizontal panel “protein, lncRNA, CpG” between panels a et b? Please enlarge the Y-axis and the p-value stars of the panel b that are unreadable. Change the Y axis scale of the methylation level to better observe the differences between the modules and add in the legend that M3 was used as reference for the statistical test? Why this choice? the profiles M12 and M3 opposite in terms of “repr” and “enh” marks and yet having apparently the same level of expression and methylation would be interesting to comment.
- Fig.3e: when you wrote “the signal intensity of H3K4me1 within EnhA clearly separated different tissue types suggesting that the signal intensity of individual epi-mark in enhancers is highly indicative of tissue identity”, what about other marks: no tissue separation?

About the tissue-specificity

- For gene identification with tissue-specific expression (TSE), different metrics have been proposed in the literature and compared (see *Bioinformatics*, 18(2), 2017, 205–214). None was used in this study. Why? You mention having performed a student test on TPM expression, which is not very appropriate. However, you referred to the method used in the paper Fang et al 2020 which is not based on a student test and which uses log transformed expressions as input. Could you clarify which statistical test was done in this paper?

- It would be interesting to illustrate the different marks and expressions (like in fig.2j) for a gene with a specific expression of a single tissue, for example the genes coding the transcription factors MEF2A, HNF1B, and HNF4A1 mentioned in fig.4 and expressed specifically in muscle, liver and intestinal tissues, respectively.

Promoter-enhancer analysis – TAD

- To evaluate how enhancers of TSE genes switch among tissues, you have generated the predicted TADs from CTCF ChIP-seq data by FIMO⁷⁰ following the method described in Oti, et al⁷¹. The identification of TAD is a broad research field. The Oti method that you published does not predict TADs but predicts CTCF loops (<https://bmcbgenomics.biomedcentral.com/articles/10.1186/s12864-016-2516-6>). Shouldn't you write "CTCF loops" instead of "TAD"? Moreover, no table providing the number and genomic localization of these TADs / CTCF loops have been provided.

Chromatin state plays an important role in pig domestication and complex traits (Fig.5)

- Could you please summarize in the text the row data provided in supp TableS8, *i.e.* how many Asian and European breeds have been analyzed and how many WGS used per breed?

Moreover, I have not seen any information about the different selection signatures that you identified. Which number and size? Please could you provide a table about them?

- Fig.5b: "above 1 dash line means significant enrichment". Could you give more information about the statistical test in method? Why the diagonal?

- Fig.5f: could you confirm that the 4 Hi-C data used comes from liver? the region seems to be not expressed in this tissue. Therefore, is it relevant to use the Hi-C data in this example focused on Muscle? Please could you clarify?

- When you wrote "in the ADG QTLs in Landrace (Fig.5e), we found that the top hit SNPs that are within a muscle-specific EnhA (Fig.5f) that appears to target two genes (ZNF532 and ALPK2)". Could you give more details in method section? Did you sequence the region to observe all the SNPs? Did you sequence enough animals to perform a GWAS on each SNP and then found that the top one is in a muscle-specific EnhA? Please could you clarify?

Heritability enrichment analysis

Could you give a few words about the "Heritability enrichment analysis" approach (ref 44), which is not common?

Reviewer #3:

Remarks to the Author:

Pan et al. 2021 "Pig genome functional annotation.."

Overall Summary: The manuscript presented by Pan et al. describes a tremendous synthesis of individual datasets, with subsequent detailed analyses providing considerable insight regarding functional annotations (regulatory elements/chromatin states) of complex traits and the dynamic epigenomic landscape (both comparatively and within the context of phylogenetics/evolution). Importantly, this study also seeks to assess and estimate which species (pig or mouse) might be more well suited for use as a model organism for human traits/disease. This is an important objective considering that the mouse-only- model dogma is just beginning to wane. To this reviewer's best knowledge, no other study of this kind exists for a domesticated livestock species. For these reasons, I think this manuscript can be published provided that some edits and clarifications occur (see below).

Edits and Clarifications

1) The manuscript contains a few rough patches in terms of language/grammar which could negatively impact readership understanding of the authors' expressions and concepts.

Lines which need attention include:

Line 73: change to "which suggests that the mouse...."

Line 92: changes to "by integrating a variety of large scale genome-wide association studies....."

Line 97-99: This is a critical sentence that is not articulated in the best way. Change to:

"....suggesting that, depending on the specific human diseases under investigation, either the pig or the mouse may be a more suitable animal model."

Line 113: need a comma after (TADs)

Line 126: You need a transition here from the preceding paragraph; otherwise it doesn't flow. Try something like this: "To illustrate the complex interplays of regulatory elements and gene expression with respect to *Escherichia coli* infection and microvillar membrane morphology in intestinal tissues, we present an analysis of Myosin 1A (MYO1A italicized)."

Line 139: Replace "Totally" with "Collectively, we identified..."

Line 140: should say (excluding Qui)

Line 142: word missing, should say "coincide with any..."

Line 147: should say "showed enrichment both up- and down-stream..."

Line 153: should say "nearby sequences" (plural)

Line 161: Same problem as above. Need a transition to ensure flow. Try this: "To explore and illustrate the relationships among chromatin states, individual epigenetic marks, gene density, gene expression, DNA methylation, and chromatin conformation we used chromosome 7 (Chr7)." Also specify in the sentence which species of chromosome 7 for clarity.

Line 163: Delete "For instance,". Begin sentence with "We observed..."

Line 165: "more physically interacted" may not be the best choice of words to deliver the intended concept of the results here.

Line 166-167: should delete "than the rest of genomic regions" and replace with something more grammatically appropriate and precise like: "within both gene desert(s) and gene rich regions than the remaining Chr7 genomic regions."

Line 168: delete "we presented", replace with "we investigated"

Line 169: delete ",as an example". This is not needed and doesn't flow well.

Line 173-174: HNF4G has not been mentioned yet in the intro or results. Therefore, placing it in this sentence without further clarification leads the reader to wonder if they've missed something. You need some sort of intro-transition to mention that gene here. Try this beginning on line 173: "These patterns were observed for MYO1A and HNF4G; a gene that....(describe relevance of HNF4G)...(Supplementary Fig. 3)."

Line 178-181: this highlights an important grammatical issue throughout the manuscript: Run-on sentences or sentences that are simply too long; where new ideas continue to be joined with "and". The entire manuscript needs to be checked with a fine-toothed comb for these and fixed with

appropriate coordinating conjunctions and/or punctuation. For instance, this sentence can be fixed by simply writing: "For instance, module 2 (M2) was characterized by active promoters and accessible enhancers; with the highest enrichment for genes and CpG islands, the lowest levels of DNA methylation, and the highest gene expression levels (Fig.3b)."

Line 193: insert "thereby" before "suggesting"

Line 213: replace "responses to wounding" with "wound responses"

Line 220-225: The understanding of this very long sentence is wounded by line 221 "whose topologically associated with tissue-specific EnhAs....". This sentence needs revision if it is to remain; especially given it's length and complexity.

Line 229-234: Example of a sentence that is too long and joined using too many instances of "and". There are 6 usages of "and"; some with commas and some not. This flows like a run-on sentence and needs to be revised to achieve a more grammatically polished statement.

Line 239-241: This sentence needs to be revised. Perhaps try: "For example, colon-specific EnhAs were associated with diseases involving recurrent bacterial infections, and cecum-specific EnhAs were significantly associated with diseases involving bruising." maybe give some examples in parenthesis too.

Line 242-244: This is a one sentence paragraph. I suggest inserting a transition statement above this (if possible), which would facilitate joining to the preceding paragraph.

Line 259: This has grammar issues and other problems. It should read "pigs are more disease resistant". But to what disease or in general? I don't want to investigate the original reference for every statement.

Line 271-273: This sentence needs to be revised to more precisely convey the authors' point.

Line 279: delete "through" and replace with "by"

Line 291: Why 50 different levels? Is it because the genomes were also segmented into 50 equal sized segments? Has 50 been used before and published? I assume there are reasons for 50 such as compute efficiency, results display efficiency etc etc.

Line 294: Not all are U shaped in Figure 6b. Should the ratio of fast vs slow be enumerated? I don't feel it's compulsory but might be interesting if the authors' agree.

Line 303-307: This sentence needs to be edited for clarity and precision. The part "genes proximal (+/-2Kb) by human-specific..." seems a bit difficult to digest/flow.

Line 333: Change to "Similar results were noted in a comparative promoter analysis (TssA)."

Line 334-335: Edit this for clarify; perhaps like this: "Our findings suggest that the pig could be a better biomedical model for certain human traits and diseases, as opposed to the mouse, and vice versa.

Line 340: Suggested edit "...across these tissues, thereby uncovering extensive..."

Line 344: There is a BMC Genomics GWAS study by Seabury et al.2017 (PMID: 28521758) on FE and Growth traits which clearly shows that positional candidate genes for these traits are functionally conserved across vertebrate species, including pigs. It may be useful to further support some of the results and discussion statements here.

Line 353-355: Again the reader needs clarification somewhere on what "more disease resistance" entails. This should be introduced earlier in the manuscript also, then revisited here.

Line 357-358: change "genomic selection programs" to plural.

Line 360: should read as "reproductive tissues"

Line 361-364: This needs to be edited for clarity and precision of the intended statement/concepts.

Line 367-372: This really should say something like: "than those which are not subject to selective pressure (i.e., the selectively neutral)." I say this because theoretical neutrality has no selective pressure. Therefore, calling it neutral selective pressure seems to be a somewhat odd choice of words, but I understand why it was initially written that way (i.e., flow).

Line 389-392: I think you can and should be a little more generous with yourselves in this statement. You can't even possibly show or discuss all your results here.

Methods Lines 550-552: "and calculated the fold enrichment of selection signature for chromatin states using the same method for gene elements enrichment described above." Where above? Can we clarify this, because this is a very large and dense paper; I'm not sure I'm looking at the correct

methods for this. A clarifying edit would help.

Reviewer Conclusions: I think the paper can and should be published after edits. I also think the style of presenting individual case-studies/examples throughout each section of the results was a wise choice, and well received. Obviously, given the breadth and scope of this work, all results cannot be described. Collectively, the methods are robust and appropriate, with few overall clarifications needed. Likewise, the figures seem clear and appropriate. Some figures are very dense, but as long as they remain high definition, one can zoom in and see everything well (main and supplemental). Supplemental Table S12 has missing ncases and ncontrols in it. This should be clarified.

-CMS

A general statement from the authors:

We thank all the reviewers for their great comments in reviewing our manuscript entitled “**Pig genome functional annotation enhances biological interpretations of complex traits and comparative epigenomics**”. Indeed, the suggestions and comments are provoke-thinking and helpful in improving the quality of our work. We have carefully considered the questions and suggestions raised by the reviewers, and revised the manuscript accordingly.

Reviewer #1 (Remarks to the Author):

I have seen this manuscript before, and this version has been significantly improved now. Generally, this work shows us a comprehensive epigenomic landscape for pig and compared the epigenomic evolution patterns among pig, mouse and human. The data resources could be used for further multi-omics research about animal evolution and livestock breeding, and scientifically shed some new insight into mammal epigenomic evolution. I'd like to recommend publication of this article after a few modifications.

Q1: The authors split the genome into 2 Mb fragments and clustered them into 12 modules in L176-L187. However, how was the fragment size determined? Is there any biological meaning like chromosome location in these modules?

Authors: The reviewer brought about a great point. Based on previous similar study by Roadmap Epigenomics Consortium ¹ (Figure 5d in that paper), in which they split the genome into 2 Mb fragments, we thus choose 2Mb. This was an arbitrary choice, which allowed us to study the overall co-occurrence of chromatin states across tissues at a relatively larger resolution (2 Mb) to recognize higher-order properties of genome. We have added this information in Line 594.

Yes, we found that some Modules were enriched in specific chromosome regions. For instance, the majority (91.7%) of Module 1 are located on the chromosome X, and chromosome Y.

Q2: In Supplementary Fig. 2, duplicates of some tissue cluster badly in the PCA analysis. The data of stomach show huge difference in almost all seq data between two replicates. As there are only 2 replicates for each tissue, this abnormal value would not be processed and infect the downstream analysis. It'd be better if the authors can explain this in text.

Authors: We agree with the reviewer that the two stomach replicates are different according to the PCA analysis. To explore whether the two stomach samples were more biologically similar compared to other tissues, we conducted the following analysis:

1. In contrast to PCA analysis, the hierarchical clustering analysis (hc.method = "complete") of all RNA-seq samples showed the two stomach samples were clustered together (Figure 1 below).
2. The expression levels of all genes between the two stomach samples were significantly correlated ($r=0.703$; $p\text{-value} < 2.2e-16$) (Figure 2a below). In addition, as the gene expression level increases, the correlation between two samples becomes higher. It reaches 0.982 when $TMM > 5$ ($n = 10,269$) (Figure 2b below).
3. The GO analysis using the tissue-specific genes identified in the two stomach samples were significantly enriched in the known biological processes related stomach function e.g., epithelial cell differentiation (GO:0030855) and smooth muscle contraction (GO:0006939) (Supplementary Fig. 8, Supplementary Table. 4).

4. Target genes of tissue-specific enhancers and promoters identified in the two stomach samples clearly showed the stomach function, e.g., smooth muscle contraction (GO:0006939), muscle contraction (GO:0006936) and response to hypoxia (GO:0001666) (Fig. 4b, Supplementary Table. 6).

In summary, the influence of including both stomach samples in the downstream analysis might be limited. We have added this information in Supplementary Fig. 4

Figure 1. Correlation of all RNA-seq samples based on TMM. Hierarchical clustering by hc.method = "complete".

Figure 2. Correlation of two RNA-seq of stomach tissue based on TMM. a, The correlation plot of two RNA-seq data from stomach tissue. b, Correlation coefficient (r) of two stomach RNA-seq data at different expression level of genes. >1 means TMM of a gene from two replicates of stomach RNA-seq over 1.

Q3: Line 124. In the figure 1e, it seems that the H3K4me1 is not significantly enriched in TSS. It looks like it is enriched at about 1kb upstream of TSS?

Authors: Thanks the reviewer for the good observation. We have modified this sentence accordingly. Overall, three active regulatory marks (ATAC, H3K4me3, H3K27ac) showed a peak at the upstream of transcription start sites (TSS) of genes across tissues (Fig. 1e), while H3K4me1 showed a peak at 1kb distance upstream of TSS (Lines 131-134).

Q4: There are too many heatmaps in main article. The authors may find more way to visualize their data efficiently and beautifully.

Authors: We modified Fig3g from heatmap to dot plot. (Please see Fig.3g)

Q5: The p value of figure 3b should have a clearer display.

Authors: Done. (Please see Fig.3b)

Q6: In figure4, why highlight the “Duodenum”?

Authors: Sorry for the confusion. As the color coding for Duodenum and Ileum looked similar, we thus highlighted ‘Duodenum’ in the original manuscript. We have removed the “highlight” and changed the color in the revised manuscript. (Please see Fig.4)

Reviewer #2 (Remarks to the Author):

Pan et al. provide the most comprehensive catalogue to date of regulatory elements in the pig by integrating 223 epigenomic and transcriptomic data sets, representing 14 biologically important tissues. They defined 15 distinct chromatin states by combining five epigenetic marks across 14 tissues. All the processed data have been integrated in the UCSC genome browser, allowing research scientists interested in specific genes to benefit from these data. Moreover, the authors provide novel biological insights at different levels (tissue specificity – selective signature – GWAS signal related to traits – conservation across species). Overall, this article is innovative and provide a useful resource for the genome functional annotation field. I have some comments and suggestions that could facilitate the reading and understanding of the manuscript, the reproducibility of the analyses and clarify some figures.

Q1: The 4 mark and ATACseq processed data (Fig.1) - It would be useful to share the bed or bigwig processed data files for these different marks.

Authors: We shared all the peak bed, processed bigwig, processed bam, chromatin state, and other processed files in http://farm.cse.ucdavis.edu/~zhypan/Nature_Communications_2021. We have added this information in Data Availability section. (Lines 812).

Q2: - Fig.1b,c, in addition to peak number and genome coverage barplots across the 14 tissues, could you also provide barplots of the different mark sizes (and their average in the text).

Authors: We have calculated the peak size for each mark of each sample and presented them in the barplot (Supplementary Fig. 2a). In addition, we provided the peak size distribution (Supplementary Fig. 2b), and added their average in the revised text (Lines 118).

Q3: - Fig.1e: The enrichment in the upstream of TSS is not as high for H3K4me1(yellow) and H3K27me3 (grey) as for the other marks which are consistent with the different chromatin states explored in the next section. Could you specify the distance used for the statistical test: 1kb? 2.5Kb? and clarify the sentence L125: “all four active regulatory marks exhibited significant enrichments in the upstream of transcription start sites (TSS) of genes”.

Authors: Sorry for the confusion. As we didn't conduct the statistical test, we modified this sentence as following:

Overall, three active regulatory marks (ATAC, H3K4me3, H3K27ac) showed a peak at the upstream of transcription start sites (TSS) of genes across tissues (Fig. 1e), while H3K4me1 showed a peak at 1kb distance upstream of TSS. (Lines 131-134)

Q4: - Fig.1f: the orientation of the MYO1A gene is missing as in the supp fig.3. It would be interesting to see the pattern across tissues for all the marks in the main fig1 or in the supp Fig.3.

Authors: We added the orientation for MYO1A gene (Fig.1f) and all the marks across tissues in supp Fig.5.

Q5: - The supp Fig.1 summarizes the data: could you please explain in the legend the nature and origin of the CTCF (16) and the Hi-C data (4) that are not a multiple of 8? i.e. what tissues and types of animals were collected for these two assays?

Authors: We have provided the details of CTCF and Hi-C data in the revised Method section. The ChIP-seq of CTCF (n = 16) from our FAANG pilot project (PRJEB14330)², including eight tissues (Adipose, Cerebellum, Cortex, Hypothalamus, Liver, Lung, Muscle, Spleen) from the two Yorkshire littermate male pigs (the same two pigs used this study). The CTCF data in the pig gut tissues were not generated. As previous studies showed TAD boundaries are conserved across tissues³ and even species^{4,5}, we thus used the CTCF (n = 16) data from these eight tissues to identify CTCF loops/TADs in this study. For the Hi-C data, the samples were collected in the liver tissue in large white pig, and the details of these data were previously described (PRJEB27364)⁶. We have added this information in the figure legend of supp Fig.1 as well.

Q6: Genome-wide chromatin state dynamics across tissues (Fig.3) - Fig.3a,b: Could you add a legend for the horizontal panel “protein, lncRNA, CpG” between panels a et b?

Authors: We have added this information in the figure legend of Fig.3. (Lines 1097-1098).

Q7: Please enlarge the Y-axis end the p-value stars of the panel b that are unreadable.

Authors: Done. (Please see Fig.3b)

Q8: Change the Y axis scale of the methylation level to better observe the differences between the modules and add in the legend that M3 was used as reference for the statistical test? Why this choice?

Authors: We modified the Y-axis scale of the methylation level (Please see Fig.3b). We chose the M3 as a reference for the statistical test, as its median methylation level (4.67) was very close to the median methylation level (4.62) of the entire genome. (Lines 602).

Q9: the profiles M12 and M3 opposite in terms of “repr” and “enh” marks and yet having apparently the same level of expression and methylation would be interesting to comment.

Authors: Thanks the reviewer for this good point! The reviewer is right. M3 and M12 had similar level of mRNA expression, but opposite directions in terms of enrichments of “repr” and “enh” marks. However, M12 had a significant lower methylation level than M3, indicating that the methylation may play an independent role in gene regulation. We have added this in the revised manuscript (Lines 224-227).

Q10: - Fig.3e: when you wrote “the signal intensity of H3K4me1 within EnhA clearly separated different tissue types suggesting that the signal intensity of individual epi-mark in enhancers is highly indicative of tissue identity”, what about other marks: no tissue separation?

Authors: Similar patterns were observed for other marks. We added these results in the Supplementary Fig.7 and line 243.

Q11: About the tissue-specificity - For gene identification with tissue-specific expression (TSE), different metrics have been proposed in the literature and compared (see Bioinformatics, 18(2), 2017, 205–214). None was used in this study. Why? You mention having perform a student test on TPM expression, which is not very appropriate. However, you referred to the method used in the paper Fang et al 2020 which is not based on a student test and which uses log transformed expressions as input. Could you clarify which statistical test was done in this paper?

Authors: The reviewer brought a valid point. We used the same method as described in Fang et al 2020⁷ and Finucane et al 2018⁸ and we have provided the detail in the revised Method section (Lines 637-645). We also mentioned that several other methods could be used to detected tissue-specific genes and cited Bioinformatics, 18(2), 2017, 205–214⁹. (Line 645)

Q12: - It would be interesting to illustrate the different marks and expressions (like in fig.2j) for a gene with a specific expression of a single tissue, for example the genes coding the transcriptions factors MEF2A, HNF1B, and HNF4A1 mentioned in fig.4 and expressed specifically in muscle, liver and intestinal tissues, respectively.

Authors: We agreed with the reviewer and checked the expression levels of MEF2A, HNF1B, HNF4A1, SIX1 and SOX10 across all studied tissues. MEF2A showed broad expression in all tissues, HNF1B showed intestine specific expression. HNF4A showed liver and intestine specific expression, while SIX1 and SOX10 showed muscle and brain specific expression, respectively. We added this information in Supplementary Fig.12 and in manuscript (Line 284).

Q13: Promoter-enhancer analysis – TAD - To evaluate how enhancers of TSE genes switch among tissues, you have generated the predicted TADs from CTCF ChIP-seq data by FIMO70 following the method described in Oti, et al⁷¹. The identification of TAD is a broad research field. The Oti method that you published does not predict TADs but predicts CTCF loops (<https://bmcbgenomics.biomedcentral.com/articles/10.1186/s12864-016-2516-6>). Shouldn't you write “CTCF loops” instead of “TAD”? Moreover, no table providing the number and genomic localization of these TADs / CTCF loops have been provided.

Authors: We agreed with the reviewer that the method described in Oti et al.¹⁰ was employed to predict CTCF loops instead of TAD. (Lines 653-667).

Briefly, the CTCF peaks from all tissues were merged, then FIMO was used to identify peaks containing the CTCF-binding motif. The directionality of the motif within peaks was used to match corresponding

boundaries of DNA loops. Nested and overlapping loops were then merged to form the predicted CTCF loops.

The predicted CTCF loops and enhancer target gene putative interacting pairs are publically available in http://farm.cse.ucdavis.edu/~zhypan/Nature_Communications_2021/Enhancer_target_gene/. We summarized these data in supp TableS5.

Q14: Chromatin state plays an important role in pig domestication and complex traits (Fig.5) - Could you please summarize in the text the row data provided in supp TableS8, i.e. how many Asian and European breeds have been analyzed and how many WGS used per breed? Moreover, I have not seen any information about the different selection signatures that you identified. Which number and size? Please could you provide a table about them?

Authors: As suggested by the reviewer, the WGS data used in the selection signature analysis has been summarized in supp TableS9.

ASD: Asian domesticated pig; ASW: Asian wild pig; EUD: European domesticated pig; EUW: European wild pig. 20 ASD breeds, 16 ASW breeds, 28 EUD breeds, 9 EUW breeds.

WGS used per breed already present in column 3 of supp TableS9.

A total of 406 whole genome sequence datasets (Supplementary Table 9) in pigs (Asian wild (58) and domestic pigs (129), European wild (35) and domestic pigs (184)) have been used to identify selection signatures. We performed Fst analysis between Asian wild and domestic pigs, and between European wild and domestic pigs with a 10-kb sliding window and 10-kb step by `popgenWindows.py` (https://github.com/simonhmartin/genomics_general). We chose the top 5% Fst regions as candidate selection signatures, and a total of 11329 selection signatures with size of 10 kb were generated. We add this information in text (Lines 701-704).

All the selection signatures are summarized in supp TableS10 and TableS11.

Q15: - Fig.5b: “above 1 dash line means significant enrichment”. Could you give more information about the statistical test in method? Why the diagonal?

Authors: We calculated the significance of enrichment using Fisher’s exact test. (Lines 706).

The diagonal line in the Figure 5b was used for distinguishing the tissues enrichment tendency either to Asian pigs or European pigs. We have clarified this in Line 1140.

Q16: - Fig.5f: could you confirm that the 4 HI-C data used comes from liver? the region seems to be not expressed in this tissue. Therefore, is it relevant to use the Hi-C data in this example focused on Muscle? Please could you clarify?

Authors: The Hi-C data used in the study were from liver, the only pig Hi-C dataset publically accessible. As previous studies have suggested the TAD boundaries are stable across tissues³ and even species^{4,5} and chromatin loops consistently form across different cell types¹¹. Therefore, Hi-C data from liver could provide evidence of potential loop interaction in muscle.

Q17: - When you wrote “in the ADG QTLs in Landrace (Fig.5e), we found that the top hit SNPs that are within a muscle-specific EnhA (Fig.5f) that appears to target two genes (ZNF532 and ALPK2) “. Could you give more details in method section? Did you sequence the region to observe all the SNPs? Did you sequence enough animals to perform a GWAS on each SNP and then found that the top one is in a muscle-specific EnhA? Please could you clarify?

Authors: We have 88,984 Landrace pigs genotyped with Genomic Profiler (GGP) Porcine LD array (8.5 K) Chip, GGP_HD_Porcine chip (43 k), Illumina PorcineSNP60 BeadChip (60 k) or GGP Porcine HD array (70 k). These genotyped animals was subjected to two-step imputation: 1) low-density markers set was imputed to HD marker set using an intermediated reference panel of 474 animals genotyped with Affymetrix Axiom PigHD SNPs chip (Axiom_PigHDv1, 658 k); 2) then the HD marker set was imputed to whole genome sequencing (WGS) level with a reference panel of 217 WGS animals. The WGS imputed animals were filtered by the following three criteria before GWAS: we filtered SNPs with 1) minor allele frequency below 0.5%, 2) with a large deviation from Hardy–Weinberg proportions ($p < 1.0 - 6$), or 3) R2 value of the imputation accuracy estimated by Minimac4 less than 0.4.

Could you give more details in method section?

Authors: Yes, more details are provided. (Lines 708-718)

Did you sequence the region to observe all the SNPs?

Authors: No, the animals for GWAS was genotyped by low-density chip and then imputed to whole genome sequencing level. Before we performed the GWAS, we filtered out SNPs with 1) minor allele frequency below 0.5%, 2) with a large deviation from Hardy–Weinberg proportions ($p < 1.0 - 6$), or 3) R2 value of the imputation accuracy estimated by Minimac4 less than 0.4.

Did you sequence enough animals to perform a GWAS on each SNP and then found that the top one is in a muscle-specific EnhA?

Authors: The number of animals used in this GWAS was 88,984. (Line 1148)

Q18: Heritability enrichment analysis Could you give a few words about the “Heritability enrichment analysis” approach (ref 44), which is not common?

Authors: Stratified linkage disequilibrium score regression (LDSC) is a commonly used approach to partition the heritability by functional annotations and to estimate the enrichment degree (i.e., the proportion of heritability explained by a functional annotation (e.g., the conserved enhancers) divided by the proportion of SNPs in this annotation) based on the GWAS summary statistics^{12,13}. It takes the population stratification factor into account through explicitly using regression modeling to quantify the relationship between linkage disequilibrium and the test statistic (χ^2 association statistic) of SNPs from GWAS, thus could improve the power and capture true polygenic signal. In this study, LDSC was used to yield the SNP-based heritability estimates, and then partition the heritability into separate functional categories to demonstrate the disproportionate contribution of different functional categories to the heritability of human complex traits and diseases. We have added this information in Lines 771-784.

Reviewer #3 (Remarks to the Author):

Pan et al. 2021 "Pig genome functional annotation.."

Overall Summary: The manuscript presented by Pan et al. describes a tremendous synthesis of individual datasets, with subsequent detailed analyses providing considerable insight regarding functional annotations (regulatory elements/chromatin states) of complex traits and the dynamic epigenomic landscape (both comparatively and within the context of phylogenics/evolution). Importantly, this study also seeks to assess and estimate which species (pig or mouse) might be more well suited for use as a model organism for human traits/disease. This is an important objective considering that the mouse-only-model dogma is just beginning to wane. To this reviewers best knowledge, no other study of this kind exists for a domesticated livestock species. For these reasons, I think this manuscript can be published

provided that some edits and clarifications occur (see below).

Edits and Clarifications

1) The manuscript contains a few rough patches in terms of language/grammar which could negatively impact readership understanding of the authors' expressions and concepts.

Lines which need attention include:

Q1: Line 73: change to "which suggests that the mouse...."

Authors: Done. (Line 75)

Q2: Line 92: changes to "by integrating a variety of large scale genome-wide association studies....."

Authors: Done. (Line 95)

Q3: Line 97-99: This is a critical sentence that is not articulated in the best way. Change to: "...suggesting that, depending on the specific human diseases under investigation, either the pig or the mouse may be a more suitable animal model."

Authors: Done. (Lines 101-102)

Q4: Line 113: need a comma after (TADs)

Authors: Done. (Line 121)

Q5: Line 126: You need a transition here from the preceding paragraph; otherwise it doesn't flow. Try something like this: "To illustrate the complex interplays of regulatory elements and gene expression with respect to *Escherichia coli* infection and microvillar membrane morphology in intestinal tissues, we present an analysis of Myosin 1A (MYO1A italicized)."

Authors: Done. (Lines 135-137)

Q6: Line 139: Replace "Totally" with "Collectively, we identified..."

Authors: Done. (Line 158)

Q7: Line 140: should say (excluding Qui)

Authors: Done. (Line 159)

Q8: Line 142: word missing, should say "coincide with any..."

Authors: Done. (Line 161)

Q9: Line 147: should say "showed enrichment both up- and down-stream..."

Authors: Done. (Line 167)

Q10: Line 153: should say "nearby sequences" (plural)

Authors: Done. (Line 179)

Q11: Line 161: Same problem as above. Need a transition to ensure flow. Try this: "To explore and illustrate the relationships among chromatin states, individual epigenetic marks, gene density, gene expression, DNA methylation, and chromatin conformation we used chromosome 7 (Chr7)." Also specify in the sentence which species of chromosome 7 for clarity.

Authors: Done. (Lines 186-188)

Q12: Line 163: Delete "For instance,". Begin sentence with "We observed..."

Authors: Done. (Line 188)

Q13: Line 165: "more physically interacted" may not be the best choice of words to deliver the intended concept of the results here.

Authors: Thank you. We modified it. (Line 190)

Q14: Line 166-167: should delete "than the rest of genomic regions" and replace with something more grammatically appropriate and precise like: "within both gene desert(s) and gene rich regions than the remaining Chr7 genomic regions."

Authors: Done. (Line 191-192)

Q15: Line 168: delete "we presented", replace with "we investigated"

Authors: Done. (Line 193)

Q16: Line 169: delete ",as an example". This is not needed and doesn't flow well.

Authors: Done. (Line 194)

Q17: Line 173-174: HNF4G has not been mentioned yet in the intro or results. Therefore, placing it in this sentence without further clarification leads the reader to wonder if they've missed something. You need some sort of intro-transition to mention that gene here. Try this beginning on line 173: "These patterns were observed for MYO1A and HNF4G; a gene that....(describe relevance of HNF4G)...(Supplementary Fig. 3)."

Authors: We have revised it as suggested. More information for this gene was added. (Lines 198-211)
These patterns were observed for *MYO1A* and Hepatocyte Nuclear Factor 4 Gamma (*HNF4G*), a gene that plays a driver role in enterocyte differentiation¹⁴ is involved renewal of intestinal stem cells¹⁵

Q18: Line 178-181: this highlights an important grammatical issue throughout the manuscript: Run-on sentences or sentences that are simply too long; where new ideas continue to be joined with "and". The entire manuscript needs to be checked with a fine-toothed comb for these and fixed with appropriate coordinating conjunctions and/or punctuation. For instance, this sentence can be fixed by simply writing: "For instance, module 2 (M2) was characterized by active promoters and accessible enhancers; with the highest enrichment for genes and CpG islands, the lowest levels of DNA methylation, and the highest gene expression levels (Fig.3b)."

Authors: We have revised them accordingly. (Lines 215-217)

Q19: Line 193: insert "thereby" before "suggesting"

Authors: Done. (Line 219)

Q20: Line 213: replace "responses to wounding" with "wound responses"

Authors: Done. (Line 264)

Q21: Line 220-225: The understanding of this very long sentence is wounded by line 221 "whose topologically associated with tissue-specific EnhAs....". This sentence needs revision if it is to remain; especially given it's length and complexity.

Authors: Done. (Lines 276-279)

Q22: Line 229-234: Example of a sentence that is too long and joined using too many instances of "and". There are 6 usages of "and"; some with commas and some not. This flows like a run-on sentence and needs to be revised to achieve a more grammatically polished statement.

Authors: Done. (Lines 284-289)

Q23: Line 239-241: This sentence needs to be revised. Perhaps try: "For example, colon-specific EnhAs were associated with diseases involving recurrent bacterial infections, and cecum-specific EnhAs were significantly associated with diseases involving bruising." maybe give some examples in parenthesis too.

Authors: Done. (Lines 311-313)

Q24: Line 242-244: This is a one sentence paragraph. I suggest inserting a transition statement above this (if possible), which would facilitate joining to the preceding paragraph.

Authors: We added a transition statement there. (Line 314)

Q25: Line 259: This has grammar issues and other problems. It should read "pigs are more disease resistant". But to what disease or in general? I don't want to investigate the original reference for every statement.

Authors: We have revised it to "more resilient". (Line 343)

Q26: Line 271-273: This sentence needs to be revised to more precisely convey the authors' point.

Authors: Done. (Lines 355-356).

Q27: Line 279: delete "through" and replace with "by"

Authors: Done. (Line 364)

Q28: Line 291: Why 50 different levels? Is it because the genomes were also segmented into 50 equal sized segments? Has 50 been used before and published? I assume there are reasons for 50 such as compute efficiency, results display efficiency etc etc.

Authors: We chose 50 different levels, in order to compare with previous findings by Xiao *et al*¹⁹, where they separated the entire genome into 50 different levels. We have this information in the Method section (Line 735).

Q29: Line 294: Not all are U shaped in Figure 6b. Should the ratio of fast vs slow be enumerated? I don't feel it's compulsory but might be interesting if the authors' agree.

Authors: Yes, this is good idea. We added following sentence for this.

However, some subtle differences among chromatin states were observed, for example, TssAHet, TxFlnkHet, and EnhAHet were in right half of the U curve, while TxFlnkWk and EnhAWK were at flat bottom of the U curve. (Lines 394-396).

Q30: Line 303-307: This sentence needs to be edited for clarity and precision. The part "genes proximal (+/-2Kb) by human-specific..." seems a bit difficult to digest/flow.

Authors: Done. (Lines 412-415).

Q31: Line 333: Change to "Similar results were noted in a comparative promoter analysis (TssA)."

Authors: Done. (Line 448)

Q32: Line 334-335: Edit this for clarify; perhaps like this: "Our findings suggest that the pig could be a better biomedical model for certain human traits and diseases, as opposed to the mouse, and vice versa.

Authors: Done. (Line 450)

Q33: Line 340: Suggested edit "...across these tissues, thereby uncovering extensive..."

Authors: Done. (Line 455)

Q34: Line 344: There is a BMC Genomics GWAS study by Seabury et al.2017 (PMID: 28521758) on FE and Growth traits which clearly shows that positional candidate genes for these traits are functionally conserved across vertebrate species, including pigs. It may be useful to further support some of the results and discussion statements here.

Authors: Thank you so much. This is helpful. We have added this information in the revision. (Lines 458-460).

Q35: Line 353-355: Again the reader needs clarification somewhere on what "more disease resistance" entails. This should be introduced earlier in the manuscript also, then revisited here.

Authors: Thanks for the reviewer's suggestion. We have revised it accordingly. (Lines 473).

Q36: Line 357-358: change "genomic selection programs" to plural.

Authors: Done. (Line 477)

Q37: Line 360: should read as "reproductive tissues"

Authors: Done. (Line 479)

Q38: Line 361-364: This needs to be edited for clarity and precision of the intended statement/concepts.

Authors: We modified this sentence. (Lines 481-482)

Q39: Line 367-372: This really should say something like: "than those which are not subject to selective pressure (i.e., the selectively neutral)." I say this because theoretical neutrality has no selective pressure. Therefore, calling it neutral selective pressure seems to be a somewhat odd choice of words, but I understand why it was initially written that way (i.e., flow).

Authors: Done. (Lines 493-494).

Q40: Line 389-392: I think you can and should be a little more generous with yourselves in this statement. You can't even possibly show or discuss all your results here.

Authors: Thanks the reviewer for your great recognition of our contribution in the field. (Line 513).

Q41: Methods Lines 550-552: "and calculated the fold enrichment of selection signature for chromatin states using the same method for gene elements enrichment described above." Where above? Can we clarify this, because this is a very large and dense paper; I'm not sure I'm looking at the correct methods for this. A clarifying edit would help.

Authors: We used the $(C/A)/(B/D)$ enrichment method:

The fold enrichment of selection signature for chromatin states using the same method for gene elements enrichment described above $(C/A)/(B/D)$, where A, B, C, D are the number of bases in a chromatin state, a gene element, overlapped between a chromatin state and a gene element, in the genome, respectively. Here the gene element was replaced with selection signature.

We added the $(C/A)/(B/D)$ method (Line 705-706).

Q42: Reviewer Conclusions: I think the paper can and should be published after edits. I also think the style of presenting individual case-studies/examples throughout each section of the results was a wise choice, and well received. Obviously, given the breadth and scope of this work, all results cannot be

described. Collectively, the methods are robust and appropriate, with few overall clarifications needed. Likewise, the figures seem clear and appropriate. Some figures are very dense, but as long as they remain high definition, one can zoom in and see everything well (main and supplemental). **Supplemental Table S12 has missing ncases and ncontrols in it.** This should be clarified.

-CMS

Authors: Some traits (such as Height, BMI and Memory because for non-binary traits) do not belong the case study, and the case and control group could not be well-defined, thus, there is no so-called case and control group in the GWASs for this types of traits. The number of cases and control are only kept for the binary traits such as Alzheimer's disease, Diverticular disease and Type 2 Diabetes. We made a note in **Supplemental Table S16.**

1. Roadmap Epigenomics, C. *et al.* Integrative analysis of 111 reference human epigenomes. *Nature* **518**, 317-330 (2015).
2. Colin Kern, Y.W., Xiaoqin Xu, Zhangyuan Pan, Michelle Halstead, Kelly Chanthavixay, Perot Saelao, Susan Waters, Ruidong Xiang, Amanda Chamberlain, Ian Korf, Mary E. Delany, Hans H. Cheng, Juan F. Medrano, Alison L. Van Eenennaam, Chris K. Tuggle, Catherine Ernst, Paul Flicek, Gerald Quon, Pablo Ross, Huaijun Zhou. Functional genome annotations of three domestic animal species provide a vital resource for comparative and agricultural research. *Nat. Commun.* **12**, 1-11 (2021).
3. Lonfat, N. & Duboule, D. Structure, function and evolution of topologically associating domains (TADs) at HOX loci. *FEBS Lett.* **589**, 2869-2876 (2015).
4. Krefting, J., Andrade-Navarro, M.A. & Ibn-Salem, J. Evolutionary stability of topologically associating domains is associated with conserved gene regulation. *BMC Biol.* **16**, 1-12 (2018).
5. Wang, M. *et al.* Putative bovine topological association domains and CTCF binding motifs can reduce the search space for causative regulatory variants of complex traits. *BMC Genomics* **19**, 1-17 (2018).
6. Foissac, S. *et al.* Multi-species annotation of transcriptome and chromatin structure in domesticated animals. *BMC Biol.* **17**, 1-25 (2019).
7. Fang, L. *et al.* Comprehensive analyses of 723 transcriptomes enhance genetic and biological interpretations for complex traits in cattle. *Genome Res.* **30**, 790-801 (2020).
8. Finucane, H.K. *et al.* Heritability enrichment of specifically expressed genes identifies disease-relevant tissues and cell types. *Nat. Genet.* **50**, 621-629 (2018).
9. Kryuchkova-Mostacci, N. & Robinson-Rechavi, M. A benchmark of gene expression tissue-specificity metrics. *Brief. Bioinform.* **18**, 205-214 (2017).
10. Oti, M., Falck, J., Huynen, M.A. & Zhou, H. CTCF-mediated chromatin loops enclose inducible gene regulatory domains. *BMC Genomics* **17**, 1-16 (2016).
11. Greenwald, W.W. *et al.* Subtle changes in chromatin loop contact propensity are associated with differential gene regulation and expression. *Nat. Commun.* **10**, 1-17 (2019).
12. Bulik-Sullivan, B.K. *et al.* LD Score regression distinguishes confounding from polygenicity in genome-wide association studies. *Nat. Genet.* **47**, 291-295 (2015).

13. Finucane, H.K. *et al.* Partitioning heritability by functional annotation using genome-wide association summary statistics. *Nat. Genet.* **47**, 1228 (2015).
14. Lindeboom, R.G. *et al.* Integrative multi-omics analysis of intestinal organoid differentiation. *Mol. Syst. Biol.* **14**, e8227 (2018).
15. Chen, L. *et al.* HNF4 regulates fatty acid oxidation and is required for renewal of intestinal stem cells in mice. *Gastroenterology* **158**, 985-999. e989 (2020).
16. Wang, L., Chan, Y. & Russell, P. *China National Commission of Animal Genetic Resources, ed. Animal Genetic Resources in China: Pigs*, (China Agriculture Press, 2011).
17. Liu, S. *et al.* Genomic analyses from non-invasive prenatal testing reveal genetic associations, patterns of viral infections, and Chinese population history. *Cell* **175**, 347-359. e314 (2018).
18. Chen, H. *et al.* Introgression of Eastern Chinese and Southern Chinese haplotypes contributes to the improvement of fertility and immunity in European modern pigs. *GigaScience* **9**, giaa014 (2020).
19. Xiao, S. *et al.* Comparative epigenomic annotation of regulatory DNA. *Cell* **149**, 1381-1392 (2012).

Reviewers' Comments:

Reviewer #1:

Remarks to the Author:

The revision has addressed most my concerns. I have no more comments.

Reviewer #2:

Remarks to the Author:

The authors have provided an improved version of the manuscript. The authors have addressed all of the comments that I had.

I have no further comments before publication.

A general statement from the authors:

We thank all the reviewers for their great work in reviewing our manuscript entitled “**Pig genome functional annotation enhances biological interpretations of complex traits and comparative epigenomics**”. Indeed, the advice and comments are provoke-thinking and helpful to improve the quality of our work.

Reviewer #1 (Remarks to the Author):

The revision has addressed most my concerns. I have no more comments.

Authors: Thank you.

Reviewer #2 (Remarks to the Author):

The authors have provided an improved version of the manuscript. The authors have addressed all of the comments that I had.

I have no further comments before publication.

Authors: Thank you.